# Multi-marginal Wasserstein GAN

**Jiezhang Cao**\*, **Langyuan Mo**\*, **Yifan Zhang, Kui Jia, Chunhua Shen, Mingkui Tan**\*†
South China University of Technology, Peng Cheng Laboratory, The University of Adelaide
{secaojiezhang, selymo, sezyifan}@mail.scut.edu.cn
{mingkuitan, kuijia}@scut.edu.cn, chunhua.shen@adelaide.edu.au

## Abstract

Multiple marginal matching problem aims at learning mappings to match a source domain to multiple target domains and it has attracted great attention in many applications, such as multi-domain image translation. However, addressing this problem has two critical challenges: (i) Measuring the multi-marginal distance among different domains is very intractable; (ii) It is very difficult to exploit cross-domain correlations to match the target domain distributions. In this paper, we propose a novel Multi-marginal Wasserstein GAN (MWGAN) to minimize Wasserstein distance among domains. Specifically, with the help of multi-marginal optimal transport theory, we develop a new adversarial objective function with inner- and inter-domain constraints to exploit cross-domain correlations. Moreover, we theoretically analyze the generalization performance of MWGAN, and empirically evaluate it on the balanced and imbalanced translation tasks. Extensive experiments on toy and real-world datasets demonstrate the effectiveness of MWGAN.

## 1 Introduction

Multiple marginal matching ($M^3$) problem aims to map an input image (source domain) to multiple target domains (see Figure 1(a)), and it has been applied in computer vision, *e.g.*, multi-domain image translation [10, 23, 25]. In practice, the unsupervised image translation [30] gains particular interest because of its label-free property. However, due to the lack of corresponding images, this task is extremely hard to learn stable mappings to match a source distribution to multiple target distributions. Recently, some methods [10, 30] address $M^3$ problem, which, however, face two main challenges.

First, existing methods often neglect to jointly optimize the multi-marginal distance among domains, which cannot guarantee the generalization performance of methods and may lead to distribution mismatching issue. Recently, CycleGAN [51] and UNIT [32] repeatedly optimize every pair of two different domains separately (see Figure 1(b)). In this sense, they are computationally expensive and may have poor generalization performance. Moreover, UFDN [30] and StarGAN [10] essentially measure the distance between an input distribution and a mixture of all target distributions (see Figure 1(b)). As a result, they may suffer from distribution mismatching issue. Therefore, it is necessary to explore a new method to measure and optimize the multi-marginal distance.

Second, it is very challenging to exploit the cross-domain correlations to match target domains. Existing methods [51, 30] only focus on the correlations between the source and target domains, since they measure the distance between two distributions (see Figure 1(b)). However, these methods often ignore the correlations among target domains, and thus they are hard to fully capture information to improve the performance. Moreover, when the source and target domains are significantly different, or the number of target domains is large, the translation task turns to be difficult for existing methods to exploit the cross-domain correlations.

---

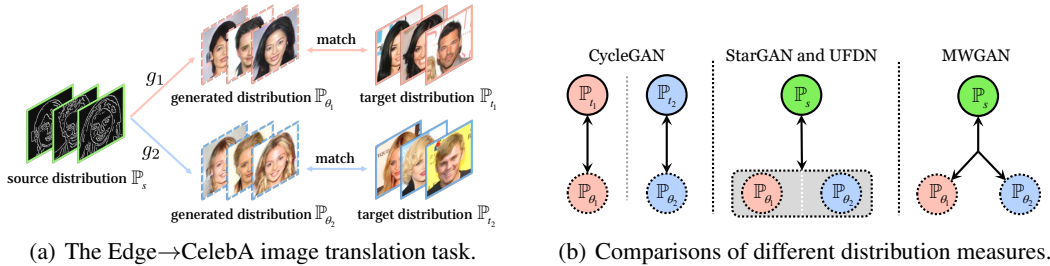

(a) The Edge→CelebA image translation task.　　(b) Comparisons of different distribution measures.

Figure 1: An example of M$^3$ problem and comparisons of existing methods. (a) For the Edge→CelebA task, we aim to learn mappings to match a source distribution (*i.e.*, Edge images) to the target distributions (*i.e.*, black and blond hair images). (b) Left: we employ CycleGAN multiple times to measure the distance between every generated distribution and its corresponding target distribution. Middle: StarGAN and UFDN measure the distance between $\mathbb{P}_s$ and a mixed distribution of $\mathbb{P}_{\theta_1}$ and $\mathbb{P}_{\theta_2}$; Right: MWGAN jointly measures Wasserstein distance among $\mathbb{P}_s$, $\mathbb{P}_{\theta_1}$ and $\mathbb{P}_{\theta_2}$. (Dotted circle: the generated distributions, solid circle: the real source or target distributions, double-headed arrow: distribution divergence, different colors represent different domains.)

In this paper, we seek to use multi-marginal Wasserstein distance to solve M$^3$ problem, but directly optimizing it is intractable. Therefore, we develop a new dual formulation to make it tractable and propose a novel multi-marginal Wasserstein GAN (MWGAN) by enforcing inner- and inter-domain constraints to exploit the correlations among domains.

The contributions of this paper are summarized as follows:

- We propose a novel GAN method (called MWGAN) to optimize a feasible multi-marginal distance among different domains. MWGAN overcomes the limitations of existing methods by alleviating the distribution mismatching issue and exploiting cross-domain correlations.

- We define and analyze the generalization of our proposed method for the multiple domain translation task, which is more important than existing generalization analyses [13, 36] studying only on two domains and non-trivial for multiple domains.

- We empirically show that MWGAN is able to solve the imbalanced image translation task well when the source and target domains are significantly different. Extensive experiments on toy and real-world datasets demonstrate the effectiveness of our proposed method.

## 2  Related Work

**Generative adversarial networks (GANs).** Deep neural networks have theoretical and experimental explorations [7, 21, 48, 49, 53]. In particular, GANs [17] have been successfully applied in computer vision tasks, such as image generation [3, 6, 18, 20], image translation [2, 10, 19] and video prediction [35]. Specifically, a generator tries to produce realistic samples, while a discriminator tries to distinguish between generated data and real data. Recently, some studies try to improve the quality [5, 9, 26] and diversity [43] of generated images, and improve the mechanism of GANs [1, 11, 38, 39] to deal with the unstable training and mode collapse problems.

**Multi-domain image translation.** M$^3$ problem can be applied in domain adaptation [45] and image translation [27, 52]. CycleGAN [51], DiscoGAN [28], DualGAN [47] and UNIT [32] are proposed to address two-domain image translation task. However, in Figure 1(b), these methods measure the distance between every pair of distributions multiple times, which is computationally expensive when applied to the multi-domain image translation task. Recently, StarGAN [10] and AttGAN [23] use a single model to perform multi-domain image translation. UFDN [30] translates images by learning domain-invariant representation for cross-domains. Essentially, the above three methods are two-domain image translation methods because they measure the distance between an input distribution and a uniform mixture of other target distributions (see Figure 1(b)). Therefore, these methods may suffer from distribution mismatching issue and obtain misleading feedback for updating models when the source and target domains are significantly different. In addition, we discuss the difference between some GAN methods in Section I in supplementary materials.

# 3 Problem Definition

**Notation.** We use calligraphic letters (*e.g.*, $\mathcal{X}$) to denote space, capital letters (*e.g.*, $X$) to denote random variables, and bold lower case letter (*e.g.*, $\mathbf{x}$) to denote the corresponding values. Let $\mathcal{D}=(\mathcal{X},\mathbb{P})$ be the domain, $\mathbb{P}$ or $\mu$ be the marginal distribution over $\mathcal{X}$ and $\mathcal{P}(\mathcal{X})$ be the set of all the probability measures over $\mathcal{X}$. For convenience, let $\mathcal{X}=\mathbb{R}^d$, and let $\mathcal{I}=\{0,...,N\}$ and $[N]=\{1,...,N\}$.

**Multiple marginal matching ($\text{M}^3$) problem.** In this paper, $\text{M}^3$ problem aims to learn mappings to match a source domain to multiple target domains. For simplicity, we consider one source domain $\mathcal{D}_s=\{\mathcal{X},\mathbb{P}_s\}$ and $N$ target domains $\mathcal{D}_i=\{\mathcal{X},\mathbb{P}_{t_i}\}, i\in[N]$, where $\mathbb{P}_s$ is the source distribution, and $\mathbb{P}_{t_i}$ is the $i$-th real target distribution. Let $g_i, i\in[N]$ be the generative models parameterized by $\theta_i$, and $\mathbb{P}_{\theta_i}$ be the generated distribution in the $i$-th target domain. In this problem, the goal is to learn multiple generative models such that each generated distribution $\mathbb{P}_{\theta_i}$ in the $i$-th target domain can be close to the corresponding real target distribution $\mathbb{P}_{t_i}$ (see Figure 1(a)).

**Optimal transport (OT) theory.** Recently, OT [42] theory has attracted great attention in many applications [3, 46]. Directly solving the primal formulation of OT [40] might be intractable [16]. To address this, we consider the dual formulation of the multi-marginal OT problem as follows.

**Problem I (Dual problem [40])** *Given $N+1$ marginals $\mu_i\in\mathcal{P}(\mathcal{X})$, potential functions $f_i, i\in\mathcal{I}$, and a cost function $c(X^{(0)},\ldots,X^{(N)}):\mathbb{R}^{d(N+1)}\to\mathbb{R}$, the dual Kantorovich problem can be defined as:*

$$W(\mu_0,...,\mu_N)=\sup_{f_i}\sum_i\int f_i\left(X^{(i)}\right)d\mu_i\left(X^{(i)}\right),\ \text{s.t.}\ \sum_i f_i\left(X^{(i)}\right)\leq c\left(X^{(0)},...,X^{(N)}\right). \quad (1)$$

In practice, we optimize the discrete case of Problem I. Specifically, given samples $\{\mathbf{x}_j^{(0)}\}_{j\in\mathcal{J}_0}$ and $\{\mathbf{x}_j^{(i)}\}_{j\in\mathcal{J}_i}$ drawn from source domain distribution $\mathbb{P}_s$ and generated target distributions $\mathbb{P}_{\theta_i}, i\in[N]$, respectively, where $\mathcal{J}_i$ is an index set and $n_i=|\mathcal{J}_i|$ is the number of samples, we have:

**Problem II (Discrete dual problem)** *Let $F=\{f_0,\ldots,f_N\}$ be the set of Kantorovich potentials, then the discrete dual problem $\hat{h}(F)$ can be defined as:*

$$\max_F \hat{h}(F)=\sum_i\frac{1}{n_i}\sum_{j\in\mathcal{J}_i}f_i\left(\mathbf{x}_j^{(i)}\right),\ \text{s.t.}\ \sum_i f_i\left(\mathbf{x}_{k_i}^{(i)}\right)\leq c\left(\mathbf{x}_{k_0}^{(0)},\ldots,\mathbf{x}_{k_N}^{(N)}\right),\forall k_i\in[n_i]. \quad (2)$$

Unfortunately, it is challenging to optimize Problem II due to the intractable inequality constraints and multiple potential functions. To address this, we seek to propose a new optimization method.

# 4 Multi-marginal Wasserstein GAN

## 4.1 A New Dual Formulation

For two domains, WGAN [3] solves Problem II by setting $f_0=f$ and $f_1=-f$. However, it is hard to extend WGAN to multiple domains. To address this, we propose a new dual formulation in order to optimize Problem II. To this end, we use a shared potential in Problem II, which is supported by empirical and theoretical evidence. In the multi-domain image translation task, the domains are often correlated, and thus share similar properties and differ only in details (see Figure 1(a)). The cross-domain correlations can be exploited by the shared potential function (see Section J in supplementary materials). More importantly, the optimal objectives of Problem II and the following problem can be equal under some conditions (see Section B in supplementary materials).

**Problem III** *Let $F_\lambda=\{\lambda_0 f,\ldots,\lambda_N f\}$ be Kantorovich potentials, then we define dual problem as:*

$$\max_{F_\lambda}\hat{h}(F_\lambda)=\sum_i\frac{\lambda_i}{n_i}\sum_{j\in\mathcal{J}_i}f\left(\mathbf{x}_j^{(i)}\right),\ \text{s.t.}\ \sum_i\lambda_i f\left(\mathbf{x}_{k_i}^{(i)}\right)\leq c\left(\mathbf{x}_{k_0}^{(0)},\ldots,\mathbf{x}_{k_N}^{(N)}\right),\forall k_i\in[n_i]. \quad (3)$$

To further build the relationship between Problem II and Problem III, we have the following theorem so that Problem III can be optimized well by GAN-based methods (see Subsection 4.2).

**Theorem 1** *Suppose the domains are connected, the cost function $c$ is continuously differentiable and each $\mu_i$ is absolutely continuous. If $(f_0,\ldots,f_N)$ and $(\lambda_0 f,\ldots,\lambda_N f)$ are solutions to Problem I, then there exist some constants $\varepsilon_i$ for each $i\in\mathcal{I}$ such that $\sum_i\varepsilon_i=0$ and $f_i=\lambda_i f+\varepsilon_i$.*

**Remark 1** *From Theorem 1, if we train a shared function $f$ to obtain a solution of Problem I, we have an equivalent Wasserstein distance, i.e., $\sum_i f_i=\sum_i\lambda_i f$ regardless of whatever the value $\varepsilon_i$ is. Therefore, we are able to optimize Problem III instead of intractable Problem II in practice.*

---

**Algorithm 1** Multi-marginal WGAN.

---

**Input:** Training data $\{\mathbf{x}_j\}_{j=1}^{n_0}$ in the initial domain, $\{\hat{\mathbf{x}}_j^{(i)}\}_{j=1}^{n_i}$ in the $i$-th target domain; batch size $m_{bs}$; the number of iterations of the discriminator per generator iteration $n_{\text{critic}}$; Uniform distribution $U[0,1]$.
**Output:** The discriminator $f$, the generators $\{g_i\}_{i\in[N]}$ and the classifier $\phi$

1: **while** not converged **do**
2:      **for** $t = 0, \ldots, n_{\text{critic}}$ **do**
3:          Sample $\mathbf{x} \sim \hat{\mathbb{P}}_s$ and $\hat{\mathbf{x}} \sim \hat{\mathbb{P}}_{\theta_i}, \forall i$, and $\tilde{\mathbf{x}} \leftarrow \rho \mathbf{x} + (1-\rho)\hat{\mathbf{x}}$, where $\rho \sim U[0,1]$
4:          Update $f$ by ascending the gradient: $\nabla_w[\mathbb{E}_{\mathbf{x}\sim\hat{\mathbb{P}}_s}[f(\mathbf{x})] - \sum_i \lambda_i^+ \mathbb{E}_{\hat{\mathbf{x}}\sim\hat{\mathbb{P}}_{\theta_i}}[f(\hat{\mathbf{x}})] - \mathcal{R}_\tau(f)]$
5:          Update classifier $\phi$ by descending the gradient $\nabla_v[\mathcal{C}_\alpha(\phi)]$
6:      **end for**
7:      Update each generator $g_i$ by descending the gradient: $\nabla_{\theta_i}[-\lambda_i^+ \mathbb{E}_{\hat{\mathbf{x}}\sim\hat{\mathbb{P}}_{\theta_i}}[f(\hat{\mathbf{x}})] - \mathcal{M}_\alpha(g_i)]$
8: **end while**

---

## 4.2 Proposed Objective Function

To minimize Wasserstein distance among domains, we now present a novel multi-marginal Wasserstein GAN (MWGAN) based on the proposed dual formulation in (3). Specifically, let $\mathcal{F} = \{f : \mathbb{R}^d \to \mathbb{R}\}$ be the class of discriminators parameterized by $w$, and $\mathcal{G} = \{g : \mathbb{R}^d \to \mathbb{R}^d\}$ be the class of generators and $g_i \in \mathcal{G}$ is parameterized by $\theta_i$. Motivated by the adversarial mechanism of WGAN, let $\lambda_0 = 1$ and $\lambda_i := -\lambda_i^+$, $\lambda_i^+ > 0, i \in [N]$, then Problem III can be rewritten as follows:

**Problem IV (Multi-marginal Wasserstein GAN)** *Given a discriminator $f \in \mathcal{F}$ and generators $g_i \in \mathcal{G}, i \in [N]$, we can define the following multi-marginal Wasserstein distance as*

$$W\left(\hat{\mathbb{P}}_s, \hat{\mathbb{P}}_{\theta_1}, \ldots, \hat{\mathbb{P}}_{\theta_N}\right) = \max_f \ \mathbb{E}_{\mathbf{x}\sim\hat{\mathbb{P}}_s}[f(\mathbf{x})] - \sum_i \lambda_i^+ \mathbb{E}_{\hat{\mathbf{x}}\sim\hat{\mathbb{P}}_{\theta_i}}[f(\hat{\mathbf{x}})], \text{ s.t. } \hat{\mathbb{P}}_{\theta_i} \in \mathcal{D}_i, f \in \Omega. \quad (4)$$

*where $\hat{\mathbb{P}}_s$ is the real source distribution, and the distribution $\hat{\mathbb{P}}_{\theta_i}$ is generated by $g_i$ in the $i$-th domain, $\Omega = \{f | f(\mathbf{x}) - \sum_{i\in[N]} \lambda_i^+ f(\hat{\mathbf{x}}^{(i)}) \leq c(\mathbf{x}, \hat{\mathbf{x}}^{(1)}, \ldots, \hat{\mathbf{x}}^{(N)}), f \in \mathcal{F}\}$ with $\mathbf{x} \in \hat{\mathbb{P}}_s$ and $\hat{\mathbf{x}}^{(i)} \in \hat{\mathbb{P}}_{\theta_i}, i \in [N]$.*

In Problem IV, we refer to $\hat{\mathbb{P}}_{\theta_i} \in \mathcal{D}_i, i \in [N]$ as **inner-domain constraints** and $f \in \Omega$ as **inter-domain constraints** (See Subsections 4.3 and 4.4). The influence of these constraints are investigated in Section N of supplementary materials. Note that $\lambda_i^+$ reflects the importance of the $i$-th target domain. In practice, we set $\lambda_i^+ = 1/N, i \in [N]$ when no prior knowledge is available on the target domains. To minimize Problem IV, we optimize the generators with the following update rule.

**Theorem 2** *If each generator $g_i \in \mathcal{G}, i \in [N]$ is locally Lipschitz (see more details of Assumption 1 [3]), then there exists a discriminator $f$ to Problem IV, we have the gradient $\nabla_{\theta_i} W(\hat{\mathbb{P}}_s, \hat{\mathbb{P}}_{\theta_1}, \ldots, \hat{\mathbb{P}}_{\theta_N}) = -\lambda_i^+ \mathbb{E}_{\mathbf{x}\sim\hat{\mathbb{P}}_s}[\nabla_{\theta_i} f(g_i(\mathbf{x}))]$ for all $\theta_i, i \in [N]$ when all terms are well-defined.*

Theorem 2 provides a good update rule for optimizing MWGAN. Specifically, we first train an optimal discriminator $f$ and then update each generator along the direction of $\mathbb{E}_{\mathbf{x}\sim\hat{\mathbb{P}}_s}[\nabla_{\theta_i} f(g_i(\mathbf{x}))]$. The detailed algorithm is shown in Algorithm 1. Specifically, the generators cooperatively exploit multi-domain correlations (see Section J in supplementary materials) and generate samples in the specific target domain to fool the discriminator; the discriminator enforces generated data in target domains to maintain the similar features from the source domain.

## 4.3 Inner-domain Constraints

In Problem IV, the distribution $\mathbb{P}_{\theta_i}$ generated by the generator $g_i$ should belong to the $i$-th domain for any $i$. To this end, we introduce an auxiliary domain classification loss and the mutual information.

**Domain classification loss.** Given an input $\mathbf{x} := \mathbf{x}^{(0)}$ and generator $g_i$, we aim to translate the input $\mathbf{x}$ to an output $\hat{\mathbf{x}}^{(i)}$ which can be classified to the target domain $\mathcal{D}_i$ correctly. To achieve this goal, we introduce an auxiliary classifier $\phi : \mathcal{X} \to \mathcal{Y}$ parameterized by $v$ to optimize the generators. Specifically, we label real data $\mathbf{x} \sim \hat{\mathbb{P}}_{t_i}$ as 1, where $\hat{\mathbb{P}}_{t_i}$ is an empirical distribution in the $i$-th target domain, and we label generated data $\hat{\mathbf{x}}^{(i)} \sim \hat{\mathbb{P}}_{\theta_i}$ as 0. Then, the domain classification loss *w.r.t.* $\phi$ can be defined as:

$$\mathcal{C}_\alpha(\phi) = \alpha \cdot \mathbb{E}_{\mathbf{x}'\sim\hat{\mathbb{P}}_{t_i}\cup\hat{\mathbb{P}}_{\theta_i}}[\ell(\phi(\mathbf{x}'), y)], \quad (5)$$

where $\alpha$ is a hyper-parameter, $y$ is corresponding to $\mathbf{x}'$, and $\ell(\cdot, \cdot)$ is a binary classification loss, such as hinge loss [50], mean square loss [34], cross-entropy loss [17] and Wasserstein loss [12].

**Mutual information maximization.** After learning the classifier $\phi$, we maximize the lower bound of the mutual information [8, 23] between the generated image and the corresponding domain, *i.e.*,

$$\mathcal{M}_\alpha(g_i) = \alpha \cdot \mathbb{E}_{\mathbf{x} \sim \hat{\mathbb{P}}_s} \left[ \log \phi \left( y^{(i)}{=}1 \,\middle|\, g_i(\mathbf{x}) \right) \right]. \tag{6}$$

By maximizing the mutual information in (6), we correlate the generated image $g_i(\mathbf{x})$ with the $i$-th domain, and then we are able to translate the source image to the specified domain.

### 4.4 Inter-domain Constraints

Then, we enforce the inter-domain constraints in Problem IV, *i.e.*, the discriminator $f \in \mathcal{F} \cap \Omega$. One can let discriminator be 1-Lipschitz continuous, but it may ignore the dependency among domains (see Section H in supplementary materials). Thus, we relax the constraints by the following lemma.

**Lemma 1 (Constraints relaxation)** *If the cost function $c(\cdot)$ is measured by $\ell_2$ norm, then there exists $L_f \geq 1$ such that the constraints in Problem IV satisfy $\sum_i |f(\mathbf{x}) - f(\hat{\mathbf{x}}^{(i)})| / \|\mathbf{x} - \hat{\mathbf{x}}^{(i)}\| \leq L_f$.*

Note that $L_f$ measures the dependency among domains (see Section G in supplementary materials). In practice, $L_f$ can be calculated with the cost function, or treated as a tuning parameter for simplicity.

**Inter-domain gradient penalty.** In practice, directly enforcing the inequality constraints in Lemma 1 would have poor performance when generated samples are far from real data. We thus propose the following inter-domain gradient penalty. Specifically, given real data $\mathbf{x}$ in the source domain and generated samples $\hat{\mathbf{x}}^{(i)}$, if $\hat{\mathbf{x}}^{(i)}$ can be properly close to $\mathbf{x}$, as suggested in [37], we can calculate its gradient and introduce the following regularization term into the objective of MWGAN, *i.e.*,

$$\mathcal{R}_\tau(f) = \tau \cdot \left( \sum_i \mathbb{E}_{\tilde{\mathbf{x}}^{(i)} \sim \hat{\mathbb{Q}}_i} \left\| \nabla f \left( \tilde{\mathbf{x}}^{(i)} \right) \right\| - L_f \right)_+^2, \tag{7}$$

where $(\cdot)_+ = \max\{0, \cdot\}$, $\tau$ is a hyper-parameter, $\tilde{\mathbf{x}}^{(i)}$ is sampled between $\mathbf{x}$ and $\hat{\mathbf{x}}^{(i)}$, and $\hat{\mathbb{Q}}_i, i \in [N]$ is a constructed distribution relying on some sampling strategy. In practice, one can construct a distribution where samples $\tilde{\mathbf{x}}^{(i)}$ can be interpolated between real data $\mathbf{x}$ and generated data $\hat{\mathbf{x}}^{(i)}$ for every domain [18]. Note that the gradient penalty captures the dependency of domains since the cost function in Problem IV measures the distance among all domains jointly.

## 5 Theoretical Analysis

In this section, we provide the generalization analysis for the proposed method. Motivated by [4], we give a new definition of generalization for multiple distributions as follows.

**Definition 1 (Generalization)** *Let $\mathbb{P}_s$ and $\mathbb{P}_{\theta_i}$ be the continuous real and generated distributions, and $\hat{\mathbb{P}}_s$ and $\hat{\mathbb{P}}_{\theta_i}$ be the empirical real and generated distributions. The distribution distance $W(\cdot, \ldots, \cdot)$ is said to generalize with $n$ training samples and error $\epsilon$, if for every true generated distribution $\mathbb{P}_{\theta_i}$, the following inequality holds with high probability,*

$$\left| W \left( \hat{\mathbb{P}}_s, \hat{\mathbb{P}}_{\theta_1}, \ldots, \hat{\mathbb{P}}_{\theta_N} \right) - W(\mathbb{P}_s, \mathbb{P}_{\theta_1}, \ldots, \mathbb{P}_{\theta_N}) \right| \leq \epsilon. \tag{8}$$

In Definition 1, the generalization bound measures the difference between the expected distance and the empirical distance. In practice, our goal is to train MWGAN to obtain a small empirical distance, so that the expected distance would also be small.

With the help of Definition 1, we are able to analyze the generalization ability of the proposed method. Let $\kappa$ be the capacity of the discriminator, and if the discriminator is $L$-Lipschitz continuous and bounded in $[-\Delta, \Delta]$, then we have the following generalization bound.

**Theorem 3 (Generalization bound)** *Given the continuous real and generated distributions $\mathbb{P}_s$ and $\mathbb{P}_{\theta_i}, i \in \mathcal{I}$, and the empirical versions $\hat{\mathbb{P}}_s$ and $\hat{\mathbb{P}}_{\theta_i}, i \in \mathcal{I}$ with at least $n$ samples in each domain, there is a universal constant $C$ such that $n \geq C \kappa \Delta^2 \log(L\kappa/\epsilon)/\epsilon^2$ with the error $\epsilon$, the following generalization bound is satisfied with probability at least $1 - e^{-\kappa}$,*

$$\left| W \left( \hat{\mathbb{P}}_s, \hat{\mathbb{P}}_{\theta_1}, \ldots, \hat{\mathbb{P}}_{\theta_N} \right) - W(\mathbb{P}_s, \mathbb{P}_{\theta_1}, \ldots, \mathbb{P}_{\theta_N}) \right| \leq \epsilon. \tag{9}$$

Theorem 3 shows that MWGAN has a good generalization ability with enough training data in each domain. In practice, if successfully minimizing the multi-domain Wasserstein distance *i.e.*, $W(\hat{\mathbb{P}}_s, \hat{\mathbb{P}}_{\theta_1}, \ldots, \hat{\mathbb{P}}_{\theta_N})$, the expected distance $W(\mathbb{P}_s, \mathbb{P}_{\theta_1}, \ldots, \mathbb{P}_{\theta_N})$ can also be small.

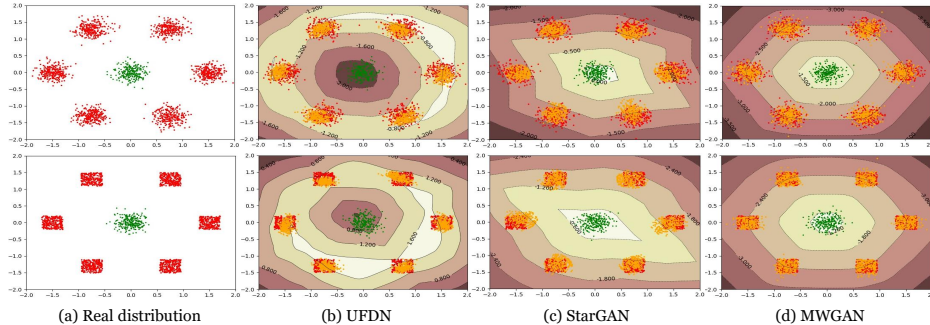

|        |        |        |        |
|--------|--------|--------|--------|
| (a) Real distribution | (b) UFDN | (c) StarGAN | (d) MWGAN |

Figure 2: Comparisons of distribution matching abilities on the value surface of discriminator. Each method learns from a Gaussian distribution to other six Gaussian (upper line) or Uniform distributions (lower line). (Green: source distribution; Red: target distributions; Orange: generated distributions. )

## 6 Experiments

**Implementation details.** All experiments are conducted based on PyTorch, with an NVIDIA TITAN X GPU.[3] We use Adam [29] with $\beta_1$=0.5 and $\beta_2$=0.999 and set the learning rate as 0.0001. We train the model 100k iterations with batch size 16. We set $\alpha$=10, $\tau$=10 and $L_f$ to be the number of target domains in Loss (7). The details of the loss function and the network architectures of the discriminator, generators and classifier can be referred to Section P in supplementary materials.

**Baselines.** We adopt the following methods as baselines: **(i) CycleGAN** [51] is a two-domain image translation method which can be flexibly extended to perform the multi-domain image translation task. **(ii) UFDN** [30] and **(iii) StarGAN** [10] are multi-domain image translation methods.

**Datasets.** We conduct experiments on three datasets. Note that all images are resized as $128 \times 128$. **(i) Toy dataset.** We generate a Gaussian distribution in the source domain, and other six Gaussian or Uniform distributions in the target domains. More details can be found in the supplemental materials. **(ii) CelebA [33]** contains 202,599 face images, where each image has 40 binary attributes. We use the following attributes: hair color (black, blond and brown), eyeglasses, mustache and pale skin. In the first experiment, we use black hair images as the source domain, and use the blond hair, eyeglasses, mustache and pale skin images as target domains. In the second experiment, we extract 50k Canny edges from CelebA. We take edge images as the source domain and hair images as target domains. **(iii) Style painting [51].** The size of Real scene, Monet, Van Gogh and Ukiyo-e is 6287, 1073, 400 and 563, respectively. We take real scene images as the source domain, and others as target domains.

**Evaluation Metrics.** We use the following evaluation metrics: **(i) Fréchet Inception Distance (FID) [24]** evaluates the quality of the translated images. In general, a lower FID score means better performance. **(ii) Classification accuracy** widely used in [10, 23] evaluates the probability that the generated images belong to corresponding target domains. Specifically, we train a classifier on CelebA (90% for training and 10% for testing) using ResNet-18 [22], resulting in a near-perfect accuracy, then use the classifier to measure the classification accuracy of the generated images.

### 6.1 Results on Toy Dataset

We compare MWGAN with UFDN and StarGAN on toy dataset to verify the limitations mentioned in Section 2. Specifically, we measure the distribution matching ability and plot the value surface of the discriminator. Here, the value surface depicts the outputs of the discriminator [18, 31].

In Figure 2, MWGAN matches the target domain distributions very well as it is able to capture the geometric information of real distribution using a low-capacity network. Moreover, the value surface shows that the discriminator provides correct gradients to update the generators. However, the baseline methods are very sensitive to the type of source and target domain distributions. With the same capacity, the baseline methods on similar distributions (top row) are able to match the target domain distributions. However, they cannot match the target domain distribution well when the initial and the target domain distributions are different (see bottom row of Figure 2).

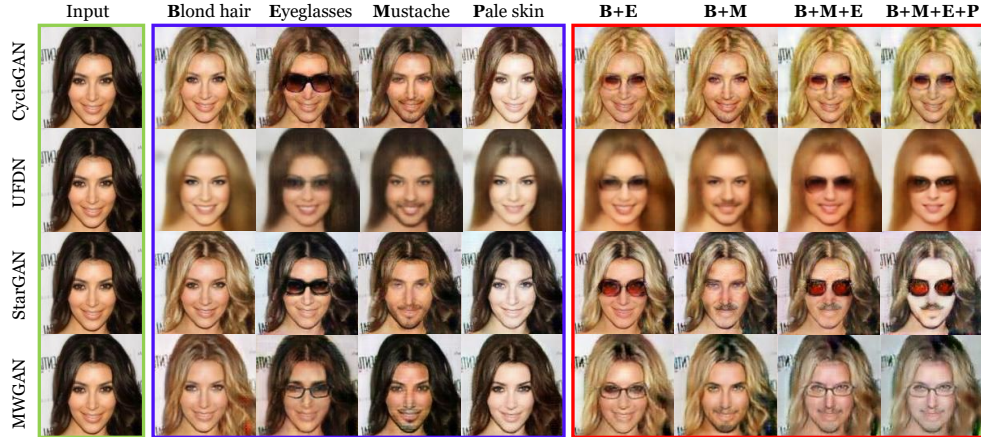

Figure 3: Comparisons of attribute translation on CelebA. The first column shows the input images, the next four columns show the single attribute translation results, and the last four columns show the multi-attribute translation results. (B: Blond hair; E: Eyeglasses; M: Mustache; P: Pale skin.)

Table 1: Comparisons of FID and classification accuracy (%) on single facial attribute translation.

| Method | Hair | | Eyeglasses | | Mustache | | Pale skin | |
|---|---|---|---|---|---|---|---|---|
| | FID | Accuracy (%) | FID | Accuracy (%) | FID | Accuracy (%) | FID | Accuracy (%) |
| CycleGAN | 20.45 | 95.07 | 23.69 | 96.94 | 24.94 | 93.89 | 18.09 | 80.75 |
| UFDN | 65.06 | 92.01 | 69.30 | 79.34 | 76.04 | 97.18 | 53.11 | 83.33 |
| StarGAN | 23.47 | 96.00 | 25.36 | 99.51 | 23.75 | **99.06** | 18.12 | 92.48 |
| MWGAN | **19.63** | **97.65** | **22.94** | **99.53** | **23.69** | 98.35 | **15.91** | **93.66** |

Table 2: Comparisons of classification accuracy (%) on multi-attribute synthesis. (B: Blond hair, E: Eyeglasses, M: Mustache, P: Pale skin.)

| Method | B+E | B+M | B+M+E | B+M+E+P |
|---|---|---|---|---|
| CycleGAN | 66.43 | 33.33 | 11.03 | 2.11 |
| UFDN | 72.53 | 51.40 | 23.00 | 8.54 |
| StarGAN | 66.66 | 62.20 | 45.77 | 6.10 |
| MWGAN | **75.82** | **69.01** | **53.75** | **19.95** |

Table 3: Comparisons of the FID value for each facial attribute (different colors of hair) on the Edge→CelebA translation task.

| Method | Black hair | Blond hair | Brown hair |
|---|---|---|---|
| CycleGAN | 65.10 | 81.59 | 65.79 |
| UFDN | 131.65 | 144.78 | 88.40 |
| StarGAN | 53.41 | 81.00 | 57.51 |
| MWGAN | **33.81** | **51.87** | **35.24** |

## 6.2    Results on CelebA

We compare MWGAN with several baselines on both balanced and imbalanced translation tasks.

*(i) Balanced image translation task.*    In this experiment, we train the generators to produce single attribute images, and then synthesize multi-attribute images using the composite generators. We generate attributes in order of {Blond hair, Eyeglasses, Mustache, Pale skin}. Taking two attributes as an example, let $g_1$ and $g_2$ be the generators of Blond hair and Eyeglasses images, respectively, then images with Blond hair and Eyeglasses attributes are generated by the composite generators $g_2 \circ g_1$.

**Qualitative results.**    In Figure 3, MWGAN has a better or comparable performance than baselines on the single attribute translation task, but achieves the highest visual quality of multi-attributes translation results. In other words, MWGAN has good generalization performance. However, CycleGAN is hard to synthesize multi-attributes. UFDN cannot guarantee the identity of the translated images and produces images with blurring structures. Moreover, StarGAN highly depends on the number of transferred domains and the synthesized images sometimes lack the perceptual realism.

**Quantitative results.** We further compare FID and classification accuracy for the single-attribute results. For the multi-attribute results, we only report classification accuracy because FID is no longer a valid measure and may give misleading results when training data are not sufficient [24]. In Table 1, MWGAN achieves the lowest FID and comparable classification accuracy, indicating that it produces realistic single-attribute images of the highest quality. In Table 2, MWGAN achieves the highest classification accuracy and thus synthesizes the most realistic multi-attribute images.

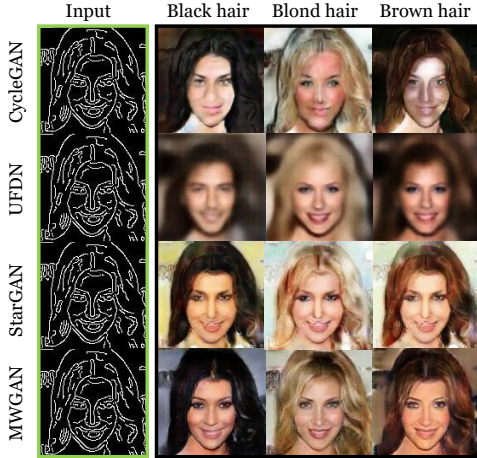
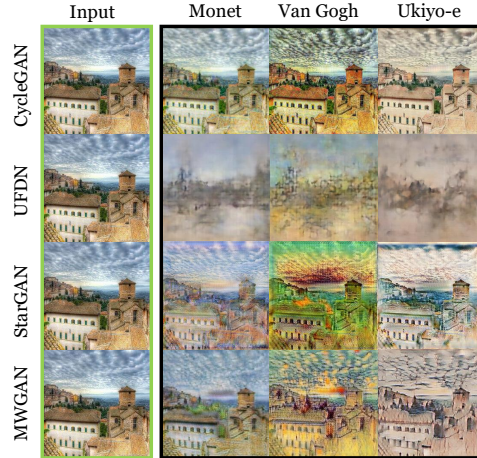

Figure 4: Comparisons of the edge→CelebA translation results. The first column shows the input images, and the next three columns show the single attribute translation results.

Figure 5: Comparisons of style transfer results. The first column shows the real world images, and the last three columns show translation results, *i.e.*, Monet, Van Gogh and Ukiyo-e.

*(ii) Imbalanced image translation task.* In this experiment, we compare MWGAN with baselines on the Edge→CelebA translation task. Note that this task is unbalanced because the information of edge images is much less than facial attribute images.

**Qualitative results.** In Figure 4, MWGAN is able to generate the most natural-looking facial images with the corresponding attributes from edge images. In contrast, UFDN fails to preserve the facial texture of an edge image, and generates images with very blurry and distorted structure. In addition, CycleGAN and StarGAN mostly preserve the domain information but cannot maintain the sharpness of images and the facial structure information. Moreover, this experiment also shows the superiority of our method on the imbalanced image translation task.

**Quantitative results.** In Table 3, MWGAN achieves the lowest FID, showing that it is able to produce the most realistic facial attributes from the edge images. In contrast, the FID values of baselines are large because these methods are hard to generate sharp and realistic images. We also perform a perceptual evaluation with AMT for this task (see Section M in supplementary materials).

### 6.3 Results on Painting Translation

In this experiment, we finally train our model on the painting dataset to conduct the style transfer task [41, 44]. As suggested in [14, 15, 51], we only show the qualitative results. Note that this translation task is also imbalanced because the input and target distributions are significantly different.

In Figure 5, MWGAN generates painting images with higher visual quality. In contrast, UFDN fails to generate clearly structural painting images because it is hard to learn domain-invariant representation when domains are highly imbalanced. CycleGAN cannot fully learn some useful information from painting images to scene images. When taking a painting image as an input, StarGAN may obtain misleading information to update the generator. In this sense, when all domains are significantly different, StarGAN may not learn a good single generator to synthesize images of multiple domains.

## 7 Conclusion

In this paper, we have proposed a novel multi-marginal Wasserstein GAN (MWGAN) for multiple marginal matching problem. Specifically, with the help of multi-marginal optimal transport theory, we develop a new dual formulation for better adversarial learning on the unsupervised multi-domain image translation task. Moreover, we theoretically define and further analyze the generalization ability of the proposed method. Extensive experiments on both toy and real-world datasets demonstrate the effectiveness of the proposed method.

**Acknowledgements**

This work is partially funded by Guangdong Provincial Scientific and Technological Funds under Grants 2018B010107001, National Natural Science Foundation of China (NSFC) 61602185, key project of NSFC (No. 61836003), Fundamental Research Funds for the Central Universities D2191240, Program for Guangdong Introducing Innovative and Enterpreneurial Teams 2017ZT07X183, and Tencent AI Lab Rhino-Bird Focused Research Program (No. JR201902). This work is also partially funded by Microsoft Research Asia (MSRA Collaborative Research Program 2019).

## Footnotes

[3]The source code of our method is available: https://github.com/caojiezhang/MWGAN.

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
