[Supplementary Material · supplementary_1034.pdf]

# Supplementary Materials: Multi-marginal Wasserstein GAN

**Jiezhang Cao**[*], **Langyuan Mo**[*], **Yifan Zhang, Kui Jia, Chunhua Shen, Mingkui Tan**[*†]
South China University of Technology, Peng Cheng Laboratory, The University of Adelaide
{secaojiezhang, selymo, sezyifan}@mail.scut.edu.cn
{mingkuitan, kuijia}@scut.edu.cn, chunhua.shen@adelaide.edu.au

**Organization.** In the supplementary materials, we provide detailed proofs for all theorems, lemmas and propositions of our paper [2], and more experiment settings and results. We organize our supplementary materials as follows.

**Theory part.** In Section A, we provide preliminaries of multi-marginal optimal transport. In Section B, we prove an equivalence theorem that solving Problem II is equivalent to solving Problem III under a mild assumption. In Section C, we build the relationship between Problem II and Problem III. In Section D, we provide an error bound of the new dual formulation. In Section E, we prove an update rule of optimizing the generators in Problem IV. In Section F, we theoretically analyze the generalization performance of MWGAN. In Section G, we provide a relaxation of inter-domain constraints. In Section H, we discuss a case that the potential function is Lipschitz continuous.

**Experiment part.** In Section I, we compare MWGAN with existing GAN methods. In Section J, we study the effectiveness of one potential function of MWGAN. In Section K, we introduce more details of toy dataset. In Section L, we introduce details of the classification on CelebA. In Section M, we apply more quantitative evaluations for MWGAN. In Section N, we discuss the influences of inner-domain constraints and inter-domain constraints. In Section O, we discuss the influence of the hyper-parameter in our proposed method. In Section P, we introduce the details about the network architecture of the discriminator and generators as well as more training details of MWGAN. In Section Q, we present more qualitative results on CelebA and style painting dataset.

## A  Preliminaries of Multi-marginal Optimal Transport

**Notation.** We use calligraphic letters (*e.g.*, $\mathcal{X}$) to denote space, capital letters (*e.g.*, $X$) to denote random variables, and bold lower case letter (*e.g.*, $\mathbf{x}$) to denote the corresponding values. Let $\mathcal{D}=(\mathcal{X}, \mathbb{P})$ be the domain, $\mathbb{P}$ or $\mu$ be the marginal distribution over $\mathcal{X}$ and $\mathcal{P}(\mathcal{X})$ be the set of all the probability measures over $\mathcal{X}$. For convenience, let $\mathcal{X}=\mathbb{R}^d$, and let $\mathcal{I}=\{0, ..., N\}$ and $[N]=\{1, ..., N\}$.

Deep learning has achieved great success in computer vision. Despite its empirical success, however, the theoretical understanding of deep neural networks still remains an open problem. Existing analysis methods [3] are hard to understand the deep neural networks. Recently, optimal transport [17, 5, 18] has been applied in deep learning [19]. With the help of optimal transport theory, one can define the following primal problem to measure the distance among all distributions jointly (see Figure 1). Specifically, the primal formulation of the multi-marginal Kantorovich problem is defined as follows.

**Problem V (Primal problem [16])** *Given* $N+1$ *marginals* $\mu_i \in \mathcal{P}(\mathbb{R}^d)$, $\forall\ i \in \mathcal{I}$ *and a cost function* $c\left(X^{(0)}, \ldots, X^{(N)}\right)$, *then the multi-marginal Kantorovich problem can be defined as:*

$$\inf_{\gamma \in \Pi(\mu_0, \ldots, \mu_N)} \int c\left(X^{(0)}, \ldots, X^{(N)}\right) d\gamma\left(X^{(0)}, \ldots, X^{(N)}\right), \tag{10}$$

*where* $\Pi(\mu_0, \ldots, \mu_N)$ *is the set of probabilistic couplings* $\gamma\left(X^{(0)}, \ldots, X^{(N)}\right)$ *with the marginal* $\mu_i$, *for all* $i \in \mathcal{I}$, $\Pi(\mu_0, \ldots, \mu_N):=\left\{\gamma \in \mathcal{P}\left(\mathbb{R}^{d(N+1)}\right) | \pi_i(\gamma)=\mu_i, \forall i \in \mathcal{I}\right\}$, *where* $\pi_i : \mathbb{R}^{d(N+1)} \rightarrow \mathbb{R}^d$ *is the canonical projection.*

---

[*]Authors contributed equally.
[†]Corresponding author.

Solving the primal problem is intractable on the generative task [8], so we consider the dual formulation of the multi-marginal Kantorovich problem.

**Problem VI (Dual problem [16])** *Given $N+1$ marginals $\mu_i \in \mathcal{P}(\mathbb{R}^d)$ and potentials $f_i, i \in \mathcal{I}$, the dual Kantorovich problem of multi-marginal Wasserstein distance is defined as:*

$$
\begin{aligned}
W(\mu_0, \ldots, \mu_N) &= \sup_{f_i} \sum_{i \in \mathcal{I}} \int f_i\left(X^{(i)}\right) d\mu_i\left(X^{(i)}\right), \\
&\text{s.t.} \sum_{i \in \mathcal{I}} f_i\left(X^{(i)}\right) \leq c\left(X^{(0)}, \ldots, X^{(N)}\right).
\end{aligned}
\tag{11}
$$

In practice, we optimize the discrete case of Problem I. Specifically, given samples $\{\mathbf{x}_j^{(0)}\}_{j \in \mathcal{J}_0}$ and $\{\mathbf{x}_j^{(i)}\}_{j \in \mathcal{J}_i}$ drawn from source domain distribution $\mathbb{P}_s$ and generated target distributions $\mathbb{P}_{\theta_i}, i \in [N]$, respectively, where $\mathcal{J}_i$ is an index set and $n_i = |\mathcal{J}_i|$ is the size of samples, we have:

**Problem VII (Discrete dual problem)** *Let $F = \{f_0, \ldots, f_N\}$ be the set of Kantorovich potentials, then the discrete dual problem $\hat{h}(F)$ can be defined as:*

$$
\max_F \ \hat{h}(F) = \sum_i \frac{1}{n_i} \sum_{j \in \mathcal{J}_i} f_i\left(\mathbf{x}_j^{(i)}\right),
\tag{12}
$$

$$
\text{s.t.} \sum_i f_i\left(\mathbf{x}_{k_i}^{(i)}\right) \leq c\left(\mathbf{x}_{k_0}^{(0)}, \ldots, \mathbf{x}_{k_N}^{(N)}\right), \forall k_i \in [n_i].
\tag{13}
$$

There is an interesting class of functions satisfying the constraint in Problem I, it is helpful for deriving Theorem 4.

**Definition 2 ($c$-conjugate function)** *Let $c : \mathbb{R}^{d(N+1)} \to \mathbb{R} \cup \{\infty\}$ be a Borel function. We say that the $(N+1)$-tuple of functions $(f_0, \ldots, f_N)$ is a c-conjugate function, $\forall i \in \mathcal{I}$, if*

$$
f_i\left(X^{(i)}\right) = \inf\left\{c\left(X^{(0)}, \ldots, X^{(N)}\right) - \sum_{j \neq i}^N f_j\left(X^{(j)}\right)\right\}.
\tag{14}
$$

With Definition 2, the following theorem builds a relationship between the primal and dual problem.

**Theorem 4 (Primal-dual Optimality [12])** *Let $(\mathbb{R}^d, \mu_0), \ldots, (\mathbb{R}^d, \mu_N)$ be Polish spaces equipped with Borel probability measures $\mu_0, \ldots, \mu_N$, then we have*

1. *There exists a solution $\gamma$ to Problem V and a c-conjugate solution $(f_0, \ldots, f_N)$ to Problem I.*

2. *The maximum value of Problem V is equal to the minimum value of Problem I.*

3. *For any solution $\gamma$ of Problem V, any c-conjugate solution of Problem I and any $(X^{(0)}, \ldots, X^{(N)})$ in the support of $\gamma$, then*

$$
\sum_{i=0}^N f_i(X^{(i)}) = c(X^{(0)}, \ldots, X^{(N)}).
$$

This Primal-dual Optimality theorem helps to derive a new dual formulation in Theorem 1.

## B  Equivalent Theorem

The Kantorovich duality theorem [16] can transform Problem II into the following problem.

**Problem VIII** *Let $F_c = \{f_1, \ldots, f_N\}$ be the set of Kantorovich potentials, then the discrete c-conjugate dual problem can be defined as:*

$$\sup_{F_c} \hat{h}(F_c) = \frac{1}{n_0} \sum_{j \in \mathcal{J}_0} f^c\left(\mathbf{x}_j^{(0)}\right) + \sum_{i=1}^{N} \frac{1}{n_i} \sum_{j \in \mathcal{J}_i} f_i\left(\mathbf{x}_j^{(i)}\right), \tag{15}$$

*where $f^c$ is the c-conjugate function defined as:*

$$f^c\left(\mathbf{x}^{(0)}\right) = \inf_{\mathbf{x}^{(1)}, \ldots, \mathbf{x}^{(N)}} \left\{ c\left(\mathbf{x}^{(0)}, \mathbf{x}^{(1)}, \ldots, \mathbf{x}^{(N)}\right) - \sum_{j=1}^{N} f_j\left(\mathbf{x}^{(j)}\right) \right\}. \tag{16}$$

**Definition 3  (Cost function)** *Given a distance function $d(\cdot, \cdot)$ which satisfies the triangle inequality, i.e., $d(\mathbf{x}, \mathbf{y}) + d(\mathbf{y}, \mathbf{z}) \geq d(\mathbf{x}, \mathbf{z}), \forall \mathbf{x}, \mathbf{y}, \mathbf{z}$, then the cost function can be defined as*

$$c\left(\mathbf{x}^{(0)}, \mathbf{x}^{(1)}, \ldots, \mathbf{x}^{(N)}\right) = \sum_{i \neq j} d\left(\mathbf{x}^{(i)}, \mathbf{x}^{(j)}\right), \quad \forall i, j \in [N]. \tag{17}$$

**Lemma 2** *Given the cost function $c(\cdot, \ldots, \cdot)$ defined in Definition 3, if $N+1$ samples $\mathbf{x}_{k_i}^{(i)} \in \mathcal{X}^{(i)}, i \in \mathcal{I}$ are overlapped, let $f_i^*, i \in [N]$ be the optimizers to Problem II, and $(f^c)^*$ be the c-conjugate function defined in Eqn. (16), then*

$$(f^c)^*\left(\mathbf{x}_{k_0}^{(0)}\right) = f_i^*\left(\mathbf{x}_{k_i}^{(i)}\right), \quad i \in [N]. \tag{18}$$

**Corollary 1** *Given the cost function $c(\cdot, \ldots, \cdot)$ defined in Definition 3, we assume $f^*$ is 1-Lipschitz continuous, the samples are bounded and the distance function satisfies $d(\mathbf{x}, \mathbf{z}) \leq d(\mathbf{x}, \mathbf{y}) + d(\mathbf{y}, \mathbf{z}) \leq Cd(\mathbf{x}, \mathbf{z})$, where $C \geq 1$, when $N+1$ samples $\mathbf{x}_{k_i}^{(i)} \in \mathcal{X}^{(i)}, i \in \mathcal{I}$ are close to each other, i.e., $c(\mathbf{x}_{k_0}^{(0)}, \ldots, \mathbf{x}_{k_N}^{(N)})$ is arbitrarily small, then $(f^c)^*(\mathbf{x}_{k_0}^{(0)})$ would be arbitrarily close to $f_i^*(\mathbf{x}_{k_i}^{(i)}), i \in [N]$, where $f_i^*, i \in [N]$ are the optimizers to Problem II, and $(f^c)^*$ is the c-conjugate function defined in Eqn. (16).*

**Lemma 3** *Suppose $f^*$ is an optimal solution to Problem III and $\sum_{i \in \mathcal{I}} \lambda_i = 0$, then $f^*$ satisfies*

$$f^*\left(\mathbf{x}^{(0)}\right) = \inf_{\mathbf{x}^{(1)}, \ldots, \mathbf{x}^{(N)}} \left\{ c\left(\mathbf{x}^{(0)}, \mathbf{x}^{(1)}, \ldots, \mathbf{x}^{(N)}\right) - \sum_{j=1}^{N} \lambda_j f^*\left(\mathbf{x}^{(j)}\right) \right\}, \quad \forall \mathbf{x} \in \mathcal{X}^0. \tag{19}$$

**Theorem 5 (Equivalent Theorem)** *Given the cost function defined in Definition 3, and $\sum_i \lambda_i = 0, i \in \mathcal{I}$ then solving Problem III is equivalent to solving Problem II, i.e., the optimal objective of Problems II and III are equal.*

## B.1 Proofs of Equivalent Theorem

**Theorem 5 (Equivalent Theorem)** *Given the cost function defined in Definition 3, and $\sum_i \lambda_i = 0, i \in \mathcal{I}$ then solving Problem III is equivalent to solving Problem II, i.e., the optimal objective of Problems II and III are equal.*

**Proof** First, we prove that any optimal solution to Problem III is a feasible solution to Problem II. Suppose that $f^*$ is the optimal solution to Problem III, from Lemma 2, we know that

$$f^* \left( \mathbf{x}^{(0)} \right) = \inf_{\mathbf{x}^{(1)}, \ldots, \mathbf{x}^{(N)}} \left\{ c \left( \mathbf{x}^{(0)}, \mathbf{x}^{(1)}, \ldots, \mathbf{x}^{(N)} \right) - \sum_{j=1}^{N} \lambda_j f^* \left( \mathbf{x}^{(j)} \right) \right\}, \quad \forall \, \mathbf{x} \in \mathcal{X}^0. \quad (20)$$

From the definition of $c$-conjugate function in Eqn. (16), we have

$$(f^*)^c \left( \mathbf{x}^{(0)} \right) = \inf_{\mathbf{x}^{(1)}, \ldots, \mathbf{x}^{(N)}} \left\{ c \left( \mathbf{x}^{(0)}, \mathbf{x}^{(1)}, \ldots, \mathbf{x}^{(N)} \right) - \sum_{j=1}^{N} \lambda_j f^* \left( \mathbf{x}^{(j)} \right) \right\}, \quad \forall \, \mathbf{x} \in \mathcal{X}^0. \quad (21)$$

Hence, $f^*$ is a feasible solution to Problem II. Therefore,

$$\hat{h}(F_c^*) \geq \hat{h}(F_\lambda^*). \quad (22)$$

Second, we prove that any optimal solution to Problem II is a feasible solution to Problem III. Suppose $f_i^*, i \in [N]$ are optimizers to Problem II. From Lemma 2, $\forall \, \mathbf{x}_{k_i}^{(i)} \in \mathcal{X}^{(i)}, i \in \mathcal{I}$ with equal value, we have $(f^c)^*(\mathbf{x}_{k_0}^{(0)}) = f_i^*(\mathbf{x}_{k_i}^{(i)}), i \in [N]$, given that the cost function satisfies the condition in Definition 3. Therefore, we can find a function $\phi$ and $\lambda_i, i \in [N]$ such that

$$\lambda_0 \phi \left( \mathbf{x}_{k_0}^{(0)} \right) = (f^c)^* \left( \mathbf{x}_{k_0}^{(0)} \right), \quad (23)$$

$$\lambda_i \phi \left( \mathbf{x}_{k_i}^{(i)} \right) = f_i^* \left( \mathbf{x}_{k_i}^{(i)} \right). \quad (24)$$

Thus, $\hat{h}(F_c^*)$ can be rewritten as

$$\hat{h}(F_c^*) = \sum_i \frac{\lambda_i}{n_i} \sum_{j \in \mathcal{J}_i} \phi \left( \mathbf{x}_j^{(i)} \right) \quad (25)$$

From the definition of $(f^c)^*$, we have

$$\sum_i \lambda_i \phi \left( \mathbf{x}^{(i)} \right) \leq c \left( \mathbf{x}^{(0)}, \ldots, \mathbf{x}^{(N)} \right), \quad (26)$$

Therefore, $\phi$ is a feasible solution to Problem III, and hence

$$\hat{h}(F_c^*) \leq \hat{h}(F_\lambda^*). \quad (27)$$

From $\hat{h}(F_c^*) \geq \hat{h}(F_\lambda^*)$ and $\hat{h}(F_c^*) \leq \hat{h}(F_\lambda^*)$, we have

$$\hat{h}(F_c^*) = \hat{h}(F_\lambda^*), \quad (28)$$

where $\hat{h}(F_\lambda^*) = \{\lambda_0 f^*, \ldots, \lambda_N f^*\}$. $\qquad \square$

## B.2 Proofs of Lemmas 2 and 3

**Lemma 2** *If the cost function $c(\cdot, \ldots, \cdot)$ satisfies Definition 3, then $\forall \mathbf{x}_{k_i}^{(i)} \in \mathcal{X}^{(i)}, i \in \mathcal{I}$, if they are equal and $f_i^*, i \in [N]$ are the optimizers to Problem II, then $(f^c)^*(\mathbf{x}_{k_0}^{(0)}) = f_i^*(\mathbf{x}_{k_i}^{(i)}), i \in [N]$, where $(f^c)^*$ is the c-conjugate function defined in Eqn. (16).*

**Proof**    We prove this by Contradiction. Without loss of generality, suppose $\forall \mathbf{x}_{k_i}^{(i)} \in \mathcal{X}^{(i)}, i \in \mathcal{I}$ are equal, and $(f^c)^*(\mathbf{x}_{k_0}^{(0)}) \neq f_i^*(\mathbf{x}_{k_i}^{(i)}), i \in [N]$. Let $\mathcal{K} = \{k_1, \ldots, k_N\}$ and $\Omega = \{\mathcal{X}_1 \times \cdots \times \mathcal{X}_N\}$. According to the definition of the $c$-conjugate function,

$$(f^c)^* \left( \mathbf{x}_{k_0}^{(0)} \right) = \inf \left\{ -\sum_{j=1}^{N} f_j^* \left( \mathbf{x}_{k_j}^{(j)} \right), \inf_{\Omega \setminus \mathcal{K}} \left\{ c \left( \mathbf{x}_{k_0}^{(0)}, \mathbf{x}^{(1)}, \ldots, \mathbf{x}^{(N)} \right) - \sum_{j=1}^{N} f_j^* \left( \mathbf{x}^{(j)} \right) \right\} \right\}.$$

For simplicity, let $\psi^*(\mathbf{x}^{(1)}, \ldots, \mathbf{x}^{(N)}) = \sum_{j=1}^{N} f_j^* \left( \mathbf{x}^{(j)} \right)$, we rewrite the above function as

$$(f^c)^* \left( \mathbf{x}_{k_0}^{(0)} \right) = \inf \left\{ -\psi^* \left( \mathbf{x}_{k_1}^{(1)}, \ldots, \mathbf{x}_{k_N}^{(N)} \right), \inf_{\Omega \setminus \mathcal{K}} \left\{ c \left( \mathbf{x}_{k_0}^{(0)}, \mathbf{x}^{(1)}, \ldots, \mathbf{x}^{(N)} \right) - \psi^* \left( \mathbf{x}^{(1)}, \ldots, \mathbf{x}^{(N)} \right) \right\} \right\}.$$

Since $(f^c)^*(\mathbf{x}_{k_0}^{(0)}) \neq f_i^*(\mathbf{x}_{k_i}^{(i)}), i \in [N]$, for any $\mathbf{x}_i^{(0)}$, we have

$$\psi^* \left( \mathbf{x}_{k_1}^{(1)}, \ldots, \mathbf{x}_{k_N}^{(N)} \right) + c \left( \mathbf{x}_i^{(0)}, \mathbf{x}_{k_1}^{(1)}, \ldots, \mathbf{x}_{k_N}^{(N)} \right)$$

$$> \inf_{\Omega \setminus \mathcal{K}} \left\{ c \left( \mathbf{x}_{k_0}^{(0)}, \mathbf{x}^{(1)}, \ldots, \mathbf{x}^{(N)} \right) - \psi^*(\mathbf{x}^{(1)}, \ldots, \mathbf{x}^{(N)}) \right\} + c \left( \mathbf{x}_i^{(0)}, \mathbf{x}_{k_1}^{(1)}, \ldots, \mathbf{x}_{k_N}^{(N)} \right)$$

$$= \inf_{\Omega \setminus \mathcal{K}} \left\{ -\psi^* \left( \mathbf{x}^{(1)}, \ldots, \mathbf{x}^{(N)} \right) + c \left( \mathbf{x}_{k_0}^{(0)}, \mathbf{x}^{(1)}, \ldots, \mathbf{x}^{(N)} \right) + c \left( \mathbf{x}_i^{(0)}, \mathbf{x}_{k_1}^{(1)}, \ldots, \mathbf{x}_{k_N}^{(N)} \right) \right\}$$

$$\geq \inf_{\Omega \setminus \mathcal{K}} \left\{ -\psi^* \left( \mathbf{x}^{(1)}, \ldots, \mathbf{x}^{(N)} \right) + c \left( \mathbf{x}_i^{(0)}, \mathbf{x}^{(1)}, \ldots, \mathbf{x}^{(N)} \right) \right\}. \tag{29}$$

Line 29 follows the fact that the definition of the cost function, $\mathbf{x}_{k_i}^{(i)} \in \mathcal{X}^{(i)}, i \in \mathcal{I}$ are equal. Suppose the number of samples in each distribution is $n$,

$$W^*(\mu_0, \ldots, \mu_N) = \sup_{F_c} \hat{h}(F_c)$$

$$= \frac{1}{n} \sum_{j \in \mathcal{J}_0} (f^c)^* \left( \mathbf{x}_j^{(0)} \right) + \sum_{i=1}^{N} \frac{1}{n} \sum_{j \in \mathcal{J}_i} f_i^* \left( \mathbf{x}_j^{(i)} \right)$$

$$= \frac{1}{n} \sum_{j \in \mathcal{J}_0} \inf_{\Omega \setminus \mathcal{K}} \left\{ c \left( \mathbf{x}_j^{(0)}, \mathbf{x}^{(1)}, \ldots, \mathbf{x}^{(N)} \right) - \psi^* \left( \mathbf{x}^{(1)}, \ldots, \mathbf{x}^{(N)} \right) \right\} + \sum_{i=1}^{N} \frac{1}{n} \sum_{j \in \mathcal{J}_i} f_i^* \left( \mathbf{x}_j^{(i)} \right)$$

$$= \frac{1}{n} \sum_{j \in \mathcal{J}_0} \inf_{\Omega \setminus \mathcal{K}} \left\{ c \left( \mathbf{x}_j^{(0)}, \mathbf{x}^{(1)}, \ldots, \mathbf{x}^{(N)} \right) - \psi^* \left( \mathbf{x}^{(1)}, \ldots, \mathbf{x}^{(N)} \right) \right\}$$

$$+ \frac{1}{n} \psi^* \left( \mathbf{x}_{k_1}^{(1)}, \ldots, \mathbf{x}_{k_N}^{(N)} \right) + \frac{1}{n} \sum_{j: \mathbf{x}_j^{(i)} \notin \mathcal{K}, i \in [N]} \psi^* \left( \mathbf{x}_j^{(1)}, \ldots, \mathbf{x}_j^{(N)} \right).$$

We can always find another function $\psi'$, such that $\psi' \left( \mathbf{x}^{(1)}, \ldots, \mathbf{x}^{(N)} \right) = \psi^* \left( \mathbf{x}^{(1)}, \ldots, \mathbf{x}^{(N)} \right)$ for $\forall \mathbf{x}^{(1)}, \ldots, \mathbf{x}^{(N)} \in \Omega \setminus \mathcal{K}$, and

$$-\psi^* \left( \mathbf{x}_{k_1}^{(1)}, \ldots, \mathbf{x}_{k_N}^{(N)} \right) > -\psi' \left( \mathbf{x}_{k_1}^{(1)}, \ldots, \mathbf{x}_{k_N}^{(N)} \right) > \inf_{\Omega \setminus \mathcal{K}} \left\{ c \left( \mathbf{x}_{k_0}^{(0)}, \mathbf{x}^{(1)}, \ldots, \mathbf{x}^{(N)} \right) - \psi^* \left( \mathbf{x}^{(1)}, \ldots, \mathbf{x}^{(N)} \right) \right\}.$$

In this case, $(f^c)' \left( \mathbf{x}_j^{(0)} \right) = (f^c)^* \left( \mathbf{x}_j^{(0)} \right), \forall j \in \mathcal{J}_0$, but

$$\psi^* \left( \mathbf{x}_{k_1}^{(1)}, \ldots, \mathbf{x}_{k_N}^{(N)} \right) < \psi' \left( \mathbf{x}_{k_1}^{(1)}, \ldots, \mathbf{x}_{k_N}^{(N)} \right).$$

Therefore, $\hat{F}_\psi > \hat{F}_c$, a contradiction. $\qquad \square$

**Corollary 1** *Given the cost function $c(\cdot, \ldots, \cdot)$ defined in Definition 3, we assume $f^*$ is 1-Lipschitz continuous, the samples are bounded and the distance function satisfies $d(\mathbf{x}, \mathbf{z}) \leq d(\mathbf{x}, \mathbf{y}) + d(\mathbf{y}, \mathbf{z}) \leq C d(\mathbf{x}, \mathbf{z})$, where $C \geq 1$, when $N+1$ samples $\mathbf{x}_{k_i}^{(i)} \in \mathcal{X}^{(i)}, i \in \mathcal{I}$ are close to each other, i.e., $c(\mathbf{x}_{k_0}^{(0)}, \ldots, \mathbf{x}_{k_N}^{(N)})$ is arbitrarily small, then $(f^c)^*(\mathbf{x}_{k_0}^{(0)})$ would be arbitrarily close to $f_i^*(\mathbf{x}_{k_i}^{(i)}), i \in [N]$, where $f_i^*, i \in [N]$ are the optimizers to Problem II, and $(f^c)^*$ is the c-conjugate function defined in Eqn. (16).*

**Proof**　We prove the case of $N=1$, it can be directly extended to the case of $N>1$. Specifically, the potential function $f_1 := -f$. Based on the definition of the cost function, we have $c(\mathbf{x}, \mathbf{y}) := d(\mathbf{x}, \mathbf{y})$. If $\mathbf{x}_{k_1}^{(1)}$ is the optimal solution, *i.e.*, $(f^c)^*\left(\mathbf{x}_{k_0}^{(0)}\right) = f^*\left(\mathbf{x}_{k_1}^{(1)}\right) + d\left(\mathbf{x}_{k_0}^{(0)}, \mathbf{x}_{k_1}^{(1)}\right)$, then

$$\left| (f^c)^*\left(\mathbf{x}_{k_0}^{(0)}\right) - f^*\left(\mathbf{x}_{k_1}^{(1)}\right) \right| = d\left(\mathbf{x}_{k_0}^{(0)}, \mathbf{x}_{k_1}^{(1)}\right). \tag{30}$$

If $\mathbf{x}_{k_1'}^{(1)}$ is the optimal solution, *i.e.*, $(f^c)^*\left(\mathbf{x}_{k_0}^{(0)}\right) = f^*\left(\mathbf{x}_{k_1'}^{(1)}\right) + d\left(\mathbf{x}_{k_0}^{(0)}, \mathbf{x}_{k_1'}^{(1)}\right)$, then

$$\left| (f^c)^*\left(\mathbf{x}_{k_0}^{(0)}\right) - f^*\left(\mathbf{x}_{k_1}^{(1)}\right) \right| = \left| f^*\left(\mathbf{x}_{k_1'}^{(1)}\right) + d\left(\mathbf{x}_{k_0}^{(0)}, \mathbf{x}_{k_1'}^{(1)}\right) - f^*\left(\mathbf{x}_{k_1}^{(1)}\right) \right| \tag{31}$$

$$\leq \left| f^*\left(\mathbf{x}_{k_1'}^{(1)}\right) - f^*\left(\mathbf{x}_{k_1}^{(1)}\right) \right| + d\left(\mathbf{x}_{k_0}^{(0)}, \mathbf{x}_{k_1'}^{(1)}\right) \tag{32}$$

$$\leq d\left(\mathbf{x}_{k_1'}^{(1)}, \mathbf{x}_{k_1}^{(1)}\right) + d\left(\mathbf{x}_{k_0}^{(0)}, \mathbf{x}_{k_1'}^{(1)}\right) \tag{33}$$

$$\leq C d\left(\mathbf{x}_{k_0}^{(0)}, \mathbf{x}_{k_1}^{(1)}\right). \tag{34}$$

For the above two cases, when $\mathbf{x}_{k_1}^{(1)}$ is close to $\mathbf{x}_{k_0}^{(0)}$, then $f^*\left(\mathbf{x}_{k_1}^{(1)}\right)$ is also close to $(f^c)^*\left(\mathbf{x}_{k_0}^{(0)}\right)$. □

**Lemma 3** *Suppose $f^*$ is an optimal solution to Problem III and $\sum_{i \in \mathcal{I}} \lambda_i = 0$, then $f^*$ satisfies*

$$f^*\left(\mathbf{x}^{(0)}\right) = \inf_{\mathbf{x}^{(1)}, \ldots, \mathbf{x}^{(N)}} \left\{ c\left(\mathbf{x}^{(0)}, \mathbf{x}^{(1)}, \ldots, \mathbf{x}^{(N)}\right) - \sum_{j=1}^{N} \lambda_j f\left(\mathbf{x}^{(j)}\right) \right\}, \quad \forall \mathbf{x} \in \mathcal{X}^0. \tag{35}$$

**Proof**　Since $f^*$ is the optimal solution to Problem III, we have

$$f^*\left(\mathbf{x}^{(0)}\right) \leq \inf_{\mathbf{x}^{(1)}, \ldots, \mathbf{x}^{(N)}} \left\{ c\left(\mathbf{x}^{(0)}, \mathbf{x}^{(1)}, \ldots, \mathbf{x}^{(N)}\right) - \sum_{j=1}^{N} \lambda_j f\left(\mathbf{x}^{(j)}\right) \right\}, \quad \forall \mathbf{x} \in \mathcal{X}^0. \tag{36}$$

We prove by contradiction. Without loss of generality, suppose there exists a $\mathbf{x}_{k_0}^{(0)}$, such that

$$f^*\left(\mathbf{x}^{(0)}\right) < \inf_{\mathbf{x}^{(1)}, \ldots, \mathbf{x}^{(N)}} \left\{ c\left(\mathbf{x}^{(0)}, \mathbf{x}^{(1)}, \ldots, \mathbf{x}^{(N)}\right) - \sum_{j=1}^{N} \lambda_j f\left(\mathbf{x}^{(j)}\right) \right\}, \quad \forall \mathbf{x} \in \mathcal{X}^0. \tag{37}$$

Note that $\mathbf{x}_{k_i}^{(i)} \in \mathcal{X}^{(i)}, i \in \mathcal{I}$ can not be equal, otherwise,

$$f^*\left(\mathbf{x}_{k_0}^{(0)}\right) = -\sum_{i \in [N]} \lambda_i f^*\left(\mathbf{x}_{k_i}^{(i)}\right) \tag{38}$$

$$= -\sum_{i \in [N]} \lambda_i f^*\left(\mathbf{x}_{k_i}^{(i)}\right) + c\left(\mathbf{x}_{k_0}^{(0)}, \mathbf{x}_{k_1}^{(1)}, \ldots, \mathbf{x}_{k_N}^{(N)}\right) \tag{39}$$

$$\geq \inf_{\mathbf{x}^{(1)}, \ldots, \mathbf{x}^{(N)}} \left\{ -\sum_{i \in [N]} \lambda_i f^*\left(\mathbf{x}^{(i)}\right) + c\left(\mathbf{x}^{(0)}, \mathbf{x}^{(1)}, \ldots, \mathbf{x}^{(N)}\right) \right\}. \tag{40}$$

It is not consistent with Eqn. (37), thus $\mathbf{x}_{k_i}^{(i)} \in \mathcal{X}^{(i)}, i \in \mathcal{I}$ can not be equal.

Therefore, there exists another function $f'$ such that $f'\left(\mathbf{x}_j^{(i)}\right) = f^*\left(\mathbf{x}_j^{(i)}\right), \forall \mathbf{x}_j^{(i)} \in \mathcal{X}^i, i \in [N]$, and $f'\left(\mathbf{x}_j^{(0)}\right) = f^*\left(\mathbf{x}_j^{(0)}\right), \forall \mathbf{x}_j^{(0)} \in \mathcal{X}^0 \setminus \mathbf{x}_{k_0}^{(0)}$ and

$$f'\left(\mathbf{x}_{k_0}^{(0)}\right) = \inf_{\mathbf{x}^{(1)}, \ldots, \mathbf{x}^{(N)}} \left\{ c\left(\mathbf{x}^{(0)}, \mathbf{x}^{(1)}, \ldots, \mathbf{x}^{(N)}\right) - \sum_{j=1}^{N} \lambda_j f\left(\mathbf{x}^{(j)}\right) \right\}.$$

It is easy to verify that $f'$ satisfies the constraints in Problem III and $\hat{h}(F_\lambda') > \hat{h}(F_\lambda^*)$, where $\hat{h}(F_\lambda') = \{\lambda_0 f', \ldots, \lambda_N f'\}$ and $\hat{h}(F_\lambda^*) = \{\lambda_0 f^*, \ldots, \lambda_N f^*\}$. Therefore, it leads to a contradiction. □

## C  Proof of Theorem 1

**Theorem 1** *Suppose the domains are connected, $c$ is continuously differentiable and that each $\mu_i$ is absolutely continuous. If $(f_0, \ldots, f_N)$ and $(\lambda_0 f, \ldots, \lambda_N f)$ are solutions to Problems I, then there exist some constant $\varepsilon_i$ for all $i \in \mathcal{I}$ such that $\sum_i \varepsilon_i = 0$, and $f_i = \lambda_i f + \varepsilon_i$.*

**Proof**   First, using a convexification trick [7], we are able to construct a $c$-conjugate solution $(f_0^c, f_1^c, \ldots, f_N^c)$ to the continuous case of Problem I, where $f_i^c$ can be defined as:

$$f_i^c\left(X^{(i)}\right) = \inf\left\{ c\left(X^{(0)}, \ldots, X^{(N)}\right) - \sum_{0 \le j < i} f_j^c\left(X^{(j)}\right) - \sum_{i < j \le N} f_j\left(X^{(j)}\right)\right\}, \quad \forall i \in \mathcal{I}. \tag{41}$$

Based on the definition of $f_i^c$ and its optimality, we have

$$f_i^c\left(X^{(i)}\right) \le \inf\left\{ c\left(X^{(0)}, \ldots, X^{(N)}\right) - \sum_{j \ne i} f_j^c\left(X^{(j)}\right)\right\}, \quad \forall i \in \mathcal{I}. \tag{42}$$

Then we iteratively obtain that $f_i\left(X^i\right) \le f_i^c\left(X^i\right)$, and using the definition of Equation (41),

$$\begin{aligned}
f_i^c\left(X^{(i)}\right) &= \inf\left\{ c\left(X^{(0)}, \ldots, X^{(N)}\right) - \sum_{0 \le j < i} f_j^c\left(X^{(j)}\right) - \sum_{i < j \le N} f_j\left(X^{(j)}\right)\right\}, \quad \forall i \in \mathcal{I} \\
&\ge \inf\left\{ c\left(X^{(0)}, \ldots, X^{(N)}\right) - \sum_{j \ne i} f_j^c\left(X^{(j)}\right)\right\}.
\end{aligned} \tag{43}$$

Combining Inequalities (42) and (43), we have a $c$-conjugate solution $(f_0^c, f_1^c, \ldots, f_N^c)$ which satisfies Definition 2. Let $\varphi_i = \lambda_i f$, $-1 \le \lambda \le 1$, using the convexification trick, we are able to find $c$-conjugate solutions $(f_0^c, \ldots, f_N^c)$ and $(\varphi_0^c, \ldots, \varphi_N^c)$ to the continuous cases of Problems II and III such that $f_i \le f_i^c$ and $\varphi_i \le \varphi_i^c$. As

$$\sum_{i \in \mathcal{I}} \int f_i\left(X^{(i)}\right) d\mu_i\left(X^{(i)}\right) = \sum_{i \in \mathcal{I}} \int f_i^c\left(X^{(i)}\right) d\mu_i\left(X^{(i)}\right),$$

we must have $f_i = f_i^c$, $\mu_i$ almost everywhere. Similarly, $\varphi_i = \varphi_i^c$, $\mu_i$ almost everywhere. We choose $X^{(i)} \in \mathcal{D}_i$ where $f_i^c$ and $\varphi_i^c$ are differentiable. Then there exist $X^{(j)}$ for all $j \ne i$ such that $\left(X^{(0)}, \ldots, X^{(i-1)}, X^{(i)}, X^{(i+1)}, \ldots, X^{(N)}\right)$ in the support of $\mu$. According to Theorem 4, we have

$$f_i^c\left(X^{(i)}\right) - c\left(X^{(0)}, \ldots, X^{(i-1)}, X^{(i)}, X^{(i+1)}, \ldots, X^{(N)}\right) = -\sum_{j \ne i} f_j^c\left(X^{(j)}\right).$$

Because $f_i^c\left(Z^{(i)}\right) - c\left(X^{(0)}, \ldots, X^{(i-1)}, Z^{(i)}, X^{(i)}, \ldots, X^{(N)}\right) \le -\sum_{j \ne i} f_j^c\left(X^{(j)}\right)$ for all other $Z^{(i)}$ we have the differential of $f_i^c$ and $c(\cdot)$ w.r.t. $X^{(i)}$ as follows

$$D_{X^{(i)}} f_i^c\left(X^{(i)}\right) = D_{X^{(i)}} c\left(X^{(0)}, \ldots, X^{(i-1)}, X^{(i)}, X^{(i+1)}, \ldots, X^{(N)}\right).$$

Similarly, we have

$$D_{X^{(i)}} \varphi_i^c\left(X^{(i)}\right) = D_{X^{(i)}} c\left(X^{(0)}, \ldots, X^{(i-1)}, X^{(i)}, X^{(i+1)}, \ldots, X^{(N)}\right).$$

Therefore, we have $D_{X^{(i)}} f_i^c\left(X^{(i)}\right) = D_{X^{(i)}} \varphi_i^c\left(X^{(i)}\right)$. As this equality holds for almost all $X^{(i)}$, we have $f_i^c\left(X^{(i)}\right) = \varphi_i^c\left(X^{(i)}\right) + \varepsilon_i$ and $f_i\left(X^{(i)}\right) = \varphi_i\left(X^{(i)}\right) + \varepsilon_i$. Choosing any $\left(X^{(0)}, \ldots, X^{(i-1)}, X^{(i)}, X^{(i+1)}, \ldots, X^{(N)}\right)$ in the support of $\gamma$, then

$$\sum_{i \in \mathcal{I}} f_i^c\left(X^{(i)}\right) = c\left(X^{(0)}, \ldots, X^{(i-1)}, X^{(i)}, X^{(i+1)}, \ldots, X^{(N)}\right) = \sum_{i \in \mathcal{I}} \varphi_i^c\left(X^{(i)}\right).$$

Therefore, $\sum_i \varepsilon_i = 0$. □

## D   Error Bound of New Dual Formulation

**Theorem 6 (Error bound)** *Suppose the function $f$ is an optimal solution to Problem III and is bounded in $[-\Delta, \Delta]$, we let $\hat{\sigma}_{k_0,\ldots,k_N} = \mathbf{1}_{[f \notin \Omega]}$, where $\Omega = \{f | \sum_i \lambda_i f(\mathbf{x}_{k_i}^{(i)}) \leq c(\mathbf{x}_{k_0}^{(0)}, \ldots, \mathbf{x}_{k_N}^{(N)})\}$ with $-1 \leq \lambda_i \leq 1$, and $\sigma = \mathbb{E}[\hat{\sigma}_{k_0,\ldots,k_N}]$ be the expectation of the probability that violates constraints in Problem III. Define $h(F_\lambda) = \sum_i \lambda_i \mathbb{E}_{\mathbf{x}^{(i)}}[f(\mathbf{x}^{(i)})]$, then the error bound between the discrete $\hat{h}$ and the continuous problem $h$ is:*

$$P\left(\left|\hat{h}(F_\lambda) - h(F_\lambda)\right| \leq \epsilon\right) > 1 - 2(N+1)\exp\left(\frac{-\underline{n}\epsilon^2}{2(N+1)^2\Delta^2}\right), \tag{44}$$

*where $\underline{n} = \min_{i \in \mathcal{I}} n_i$. Let $M = \prod_{i \in \mathcal{I}} n_i$, then we have $P(|\sigma| > \epsilon) \leq 2e^{-2M\epsilon^2}$.*

**Proof**   Based on the definitions of $\hat{h}(F_\lambda)$ and $h(F_\lambda)$ and $-1 \leq \lambda_i \leq 1, i \in \mathcal{I}$,

$$\left|\hat{h}(F_\lambda) - h(F_\lambda)\right| = \left|\sum_i \frac{\lambda_i}{n_i}\sum_{j \in \mathcal{J}_i} f\left(\mathbf{x}_j^{(i)}\right) - \sum_i \lambda_i \mathbb{E}\left[f\left(\mathbf{x}^{(i)}\right)\right]\right|$$

$$= \left|\sum_i \lambda_i \left(\frac{1}{n_i}\sum_{j \in \mathcal{J}_i} f\left(\mathbf{x}_j^{(i)}\right) - \mathbb{E}\left[f\left(\mathbf{x}^{(i)}\right)\right]\right)\right|, -1 \leq \lambda_i \leq 1$$

$$\leq \sum_i \left|\frac{1}{n_i}\sum_{j \in \mathcal{J}_i} f\left(\mathbf{x}_j^{(i)}\right) - \mathbb{E}\left[f\left(\mathbf{x}^{(i)}\right)\right]\right|. \tag{45}$$

Suppose the function $f$ is bounded in $[-\Delta, \Delta]$, then, according to Hoeffding's inequality, we have

$$P\left(\left|\frac{1}{n_i}\sum_j f\left(\mathbf{x}_j^{(i)}\right) - \mathbb{E}\left[f\left(\mathbf{x}^{(i)}\right)\right]\right| > \frac{\epsilon}{N+1}\right) \leq 2\exp\left(\frac{-n_i\epsilon^2}{2(N+1)^2\Delta^2}\right). \tag{46}$$

Using Inequality (46) and union bound over all $i \in \mathcal{I}$, we further have the following inequality,

$$P\left(\bigcup_{i \in \mathcal{I}}\left(\left|\frac{1}{n_i}\sum_j f\left(\mathbf{x}_j^{(i)}\right) - \mathbb{E}\left[f\left(\mathbf{x}^{(i)}\right)\right]\right| > \frac{\epsilon}{N+1}\right)\right)$$

$$\leq \sum_i P\left(\left|\frac{1}{n_i}\sum_j f\left(\mathbf{x}_j^{(i)}\right) - \mathbb{E}\left[f\left(\mathbf{x}^{(i)}\right)\right]\right| > \frac{\epsilon}{N+1}\right)$$

$$\leq 2(N+1)\exp\left(\frac{-\underline{n}\epsilon^2}{2(N+1)^2\Delta^2}\right),$$

where $\underline{n} = \min_{i \in \mathcal{I}} n_i$. Equivalently, we rewrite the above inequality as

$$P\left(\bigcap_{i \in \mathcal{I}}\left(\left|\frac{1}{n_i}\sum_j f\left(\mathbf{x}_j^{(i)}\right) - \mathbb{E}\left[f\left(\mathbf{x}^{(i)}\right)\right]\right| \leq \frac{\epsilon}{N+1}\right)\right) > 1 - 2(N+1)\exp\left(\frac{-\underline{n}\epsilon^2}{2(N+1)^2\Delta^2}\right).$$

Therefore, from Inequality (45), the following probability inequality satisfies:

$$P\left(\left|\hat{h}(F_\lambda) - h(F_\lambda)\right| \leq \epsilon\right) \geq P\left(\sum_i \left|\frac{1}{n_i}\sum_j f\left(\mathbf{x}_j^{(i)}\right) - \mathbb{E}\left[f\left(\mathbf{x}^{(i)}\right)\right]\right| \leq \epsilon\right)$$

$$> 1 - 2(N+1)\exp\left(\frac{-\underline{n}\epsilon^2}{2(N+1)^2\Delta^2}\right).$$

Based on the definitions of $\hat{\sigma}_{k_0,\ldots,k_N}$ and $\sigma$, they are bounded in the interval $[0,1]$. Using Hoeffding's inequality, we have

$$P\left(\left|\frac{1}{M}\sum_{k_0,\ldots,k_N}\hat{\sigma}_{k_0,\ldots,k_N} - \sigma\right| > \epsilon\right) \leq 2\exp\left(-2M\epsilon^2\right),$$

where $M = \prod_i n_i$. Suppose the function $f$ can be learned by a deep neural network with sufficient capacity, and it is able to solve Problem III. Then, the inequality constraints $\sum_i f\left(\mathbf{x}_{k_i}^{(i)}\right) \leq c\left(\mathbf{x}_{k_0}^{(0)}, \ldots, \mathbf{x}_{k_N}^{(N)}\right)$ are satisfied, and thus $\hat{\sigma}_{k_0,\ldots,k_N} = 0$ for $\forall\, k_0, \ldots, k_N$, we have

$$P(|\sigma| > \epsilon) \leq 2e^{-2M\epsilon^2}.$$

$\square$

# E Proof of Theorem 2

**Definition 4** **(Function space)** *Let $\mathcal{X} \subseteq \mathbb{R}^d$ be a compact set (such as $[0,1]^d$ the space of images), the function space can be defined as*

$$C_b(\mathcal{X}) = \{f : \mathcal{X} \to \mathbb{R}, f \text{ is continuous and bounded}\}. \tag{47}$$

**Assumption 1** *[1] Let $g : \mathcal{X} \to \mathcal{X}$ be locally Lipschitz between finite dimensional vector spaces. Given $g_\theta(X)$ evaluated on coordinates $(X, \theta)$, we say that $g$ satisfies assumption 1 for a certain probability distribution $p$ over $\mathcal{X}$ if there are local Lipschitz constants $L(\theta, \mathbf{x})$ such that*

$$\mathbb{E}_{x \sim \mathbb{P}}[L(\theta, \mathbf{x})] < +\infty.$$

**Assumption 2** *[1] Assume the discriminator $f$ is Lipschitz continuous w.r.t. $\mathbf{x}$.*

**Lemma 4** *[1] Assume the discriminator $f$ is 1-Lipschitz w.r.t. $w$ and the generator $g_\theta(x)$ is locally Lipschitz as a function of $(\theta, x)$, then $\nabla_\theta \mathbb{E}_{\mathbf{x} \sim \mathbb{P}_s}[f(g_\theta(\mathbf{x}))] = \mathbb{E}_{\mathbf{x} \sim \mathbb{P}_s}[\nabla_\theta f(g_\theta(\mathbf{x}))]$.*

**Theorem 2** *If each generator $g_i \in \mathcal{G}, i \in [N]$ is locally Lipschitz and satisfies Assumption 1 [1] , then there exists a discriminator $f$ to Problem IV, we have the gradient $\nabla_{\theta_i} W(\hat{\mathbb{P}}_s, \hat{\mathbb{P}}_{\theta_1}, \ldots, \hat{\mathbb{P}}_{\theta_N}) = -\lambda_i^+ \mathbb{E}_{\mathbf{x} \sim \hat{\mathbb{P}}_s}[\nabla_{\theta_i} f(g_i(\mathbf{x}))]$ for all $\theta_i, i \in [N]$ when all terms are well-defined.*

**Proof** Recall the optimization problem, we first define the value function as follows:

$$
\begin{aligned}
V(\tilde{f}, \theta) &= \mathbb{E}_{\mathbf{x} \sim \mathbb{P}_s}\left[\tilde{f}(\mathbf{x})\right] - \frac{1}{N} \sum_i \mathbb{E}_{\mathbf{x} \sim \mathbb{P}_{\theta_i}}\left[\tilde{f}(\mathbf{x})\right] \\
&= \mathbb{E}_{\mathbf{x} \sim \mathbb{P}_s}\left[\tilde{f}(\mathbf{x})\right] - \frac{1}{N} \sum_i \mathbb{E}_{\mathbf{x} \sim \mathbb{P}_s}\left[\tilde{f}(g_i(\mathbf{x}))\right],
\end{aligned}
$$

where $\theta$ is the set of $\theta_i, i \in [N]$, $\tilde{f}$ lies in $\tilde{\mathcal{F}} = \{\tilde{f} : \mathcal{X} \to \mathbb{R}, \tilde{f} \in C_b(\mathcal{X}), \tilde{f} \in \Omega\}$ and $C_b(\mathcal{X})$ is defined in (47).

$$
\Omega = \left\{ \tilde{f} : \tilde{f}(\mathbf{x}) - \frac{1}{N} \sum_{i \in [N]} \tilde{f}\left(\mathbf{x}^{(i)}\right) \le c\left(\mathbf{x}^{(0)}, \mathbf{x}^{(1)}, \ldots, \mathbf{x}^{(N)}\right) \right\},
$$

where $\mathbf{x}^{(0)} := \mathbf{x}$. Since $\mathcal{X}$ is compact, and based on Theorem 4, there is a solution $f \in \tilde{\mathcal{F}}$ that satisfies

$$W(\mathbb{P}_s, \mathbb{P}_{\theta_1}, \ldots, \mathbb{P}_{\theta_N}) = \sup_{\tilde{f} \in \tilde{\mathcal{F}}} V(\tilde{f}, \theta) = V(f, \theta).$$

Define the optimal set $\mathcal{F}^*(\theta) = \{f \in \tilde{\mathcal{F}} : V(f, \theta) = W(\mathbb{P}_s, \mathbb{P}_{\theta_1}, \ldots, \mathbb{P}_{\theta_N})\}$, and note that this set $\mathcal{F}^*(\theta)$ is non-empty. Based on envelope theorem [15], we have

$$\nabla_{\theta_i} W(\mathbb{P}_s, \mathbb{P}_{\theta_1}, \ldots, \mathbb{P}_{\theta_N}) = \nabla_{\theta_i} V(f, \theta)$$

for any $f \in \mathcal{F}^*(\theta)$ when all terms are well-defined. Note that $f$ exists since $\mathcal{F}^*(\theta)$ is non-empty for all $\theta_i$. Then, we have

$$
\begin{aligned}
\nabla_{\theta_i} W(\mathbb{P}_s, \mathbb{P}_{\theta_1}, \ldots, \mathbb{P}_{\theta_N}) &= \nabla_{\theta_i} V(f, \theta) \\
&= \nabla_{\theta_i}\left[\mathbb{E}_{\mathbf{x} \sim \mathbb{P}_s}[f(\mathbf{x})] - \frac{1}{N} \sum_i \mathbb{E}_{\mathbf{x} \sim \mathbb{P}_s}[f(g_i(\mathbf{x}))]\right] \\
&= \nabla_{\theta_i}\left[\mathbb{E}_{\mathbf{x} \sim \mathbb{P}_s}[f(\mathbf{x})] - \frac{1}{N} \mathbb{E}_{\mathbf{x} \sim \mathbb{P}_s}[f(g_i(\mathbf{x}))]\right] \\
&= -\frac{1}{N} \nabla_{\theta_i} \mathbb{E}_{\mathbf{x} \sim \mathbb{P}_s}[f(g_i(\mathbf{x}))] \\
&= -\frac{1}{N} \mathbb{E}_{\mathbf{x} \sim \mathbb{P}_s}[\nabla_{\theta_i} f(g_i(\mathbf{x}))],
\end{aligned}
$$

where the last equality holds by Lemma 4. $\qquad\square$

# F   Proof of Theorem 3

**Theorem 3 (Generalization bound)**   *Let $\mathbb{P}_s$ and $\mathbb{P}_{\theta_i}$ be the continuous real and generated distributions, and $\hat{\mathbb{P}}_s$ and $\hat{\mathbb{P}}_{\theta_i}$ be the empirical real and generated distributions with at least $n$ samples each. When $n \geq \frac{C\kappa\Delta^2 \log(L\kappa/\epsilon)}{\epsilon^2}$, the following generalization bound is satisfied with probability at least $1 - e^{-\kappa}$,*

$$\left| W\left( \hat{\mathbb{P}}_s, \hat{\mathbb{P}}_{\theta_1}, \ldots, \hat{\mathbb{P}}_{\theta_N} \right) - W(\mathbb{P}_s, \mathbb{P}_{\theta_1}, \ldots, \mathbb{P}_{\theta_N}) \right| \leq \epsilon.$$

**Proof**   Let $\tilde{\mathcal{W}}$ be a finite set such that every point $w \in \mathcal{W}$ is within distance $\frac{\epsilon}{8L}$ of a point $w' \in \tilde{\mathcal{W}}$, *i.e.*, for every $w \in \mathcal{W}$, there exist a $w' \in \mathcal{W}$ such that $\|w - w'\| \leq \frac{\epsilon}{8L}$. For any $\mathbf{x} \in \mathbb{P}_s$ or $\mathbf{x} \in \hat{\mathbb{P}}_s$, assume that $f$ is $L$-Lipschitz continuous *w.r.t.* $w$, then we have

$$|f_{w'}(\mathbf{x}) - f_w(\mathbf{x})| \leq L\|w' - w\| \leq \frac{\epsilon}{8}. \tag{48}$$

Assume that $f$ is bounded in $[-\Delta, \Delta]$. Using to Hoeffding's inequality, for every $w' \in \tilde{\mathcal{W}}$, we have

$$P\left( \left| \mathbb{E}_{\mathbf{x}\sim\mathbb{P}_s} [f_{w'}(\mathbf{x})] - \mathbb{E}_{\mathbf{x}\sim\hat{\mathbb{P}}_s} [f_{w'}(\mathbf{x})] \right| \geq \frac{\epsilon}{4} \right) \leq 2\exp\left( -\frac{n\epsilon^2}{32\Delta^2} \right).$$

Therefore, when $n \geq \frac{C\kappa\Delta^2 \log(L\kappa/\epsilon)}{\epsilon^2}$ for a large enough constant $C$, we have union bounds over all $w' \in \tilde{\mathcal{W}}$. Then, we have $|\mathbb{E}_{\mathbf{x}\sim\mathbb{P}_s} f_{w'}(\mathbf{x}) - \mathbb{E}_{\mathbf{x}\sim\hat{\mathbb{P}}_s} f_{w'}(\mathbf{x})| \leq \frac{\epsilon}{4}$ with the high probability at least $1 - \exp(-\kappa)$, where $\kappa$ is the number of parameters in the discriminator $f$. For every $w \in \mathcal{W}$, we can find a $w' \in \tilde{\mathcal{W}}$ such that the following satisfies

$$\left| \mathbb{E}_{\mathbf{x}\sim\mathbb{P}_s} [f_w(\mathbf{x})] - \mathbb{E}_{\mathbf{x}\sim\hat{\mathbb{P}}_s} [f_w(\mathbf{x})] \right| \leq \left| \mathbb{E}_{\mathbf{x}\sim\mathbb{P}_s} [f_{w'}(\mathbf{x})] - \mathbb{E}_{\mathbf{x}\sim\mathbb{P}_s} [f_w(\mathbf{x})] \right| + \left| \mathbb{E}_{\mathbf{x}\sim\hat{\mathbb{P}}_s} [f_{w'}(\mathbf{x})] - \mathbb{E}_{\mathbf{x}\sim\hat{\mathbb{P}}_s} [f_w(\mathbf{x})] \right|$$
$$+ \left| \mathbb{E}_{\mathbf{x}\sim\mathbb{P}_s} [f_{w'}(\mathbf{x})] - \mathbb{E}_{\mathbf{x}\sim\hat{\mathbb{P}}_s} [f_{w'}(\mathbf{x})] \right|$$
$$\leq \frac{\epsilon}{8} + \frac{\epsilon}{8} + \frac{\epsilon}{4} \leq \frac{\epsilon}{2}.$$

The third line holds by Inequality (48). Therefore, with high probability at least $1 - \exp(-\kappa)$, for every discriminator $f_w$,

$$\left| \mathbb{E}_{\mathbf{x}\sim\mathbb{P}_s} [f_w(\mathbf{x})] - \mathbb{E}_{\mathbf{x}\sim\hat{\mathbb{P}}_s} [f_w(\mathbf{x})] \right| \leq \frac{\epsilon}{2}.$$

Similarly, for $\mathbb{P}_{\theta_i}$ and $\hat{\mathbb{P}}_{\theta_i}$, when $n \geq \frac{C\kappa\Delta^2 \log(L\kappa/\epsilon)}{\epsilon^2}$, with the probability at least $1 - \exp(-\kappa)$,

$$\left| \mathbb{E}_{\mathbf{x}\sim\mathbb{P}_{\theta_i}} [f_w(\mathbf{x})] - \mathbb{E}_{\mathbf{x}\sim\hat{\mathbb{P}}_{\theta_i}} [f_w(\mathbf{x})] \right| \leq \frac{\epsilon}{2}, \quad \forall i = 1, \ldots, N.$$

Let $f_w$ be the optimal discriminator of $W(\mathbb{P}_s, \mathbb{P}_{\theta_1}, \ldots, \mathbb{P}_{\theta_N})$, we have

$$W\left( \hat{\mathbb{P}}_s, \hat{\mathbb{P}}_{\theta_1}, \ldots, \hat{\mathbb{P}}_{\theta_N} \right) = \sup_{f\in\mathcal{F}} \mathbb{E}_{\mathbf{x}\sim\hat{\mathbb{P}}_s} [f(\mathbf{x})] - \frac{1}{N}\sum_i \left[ \mathbb{E}_{\mathbf{x}\sim\hat{\mathbb{P}}_{\theta_i}} [f(\mathbf{x})] \right]$$

$$\geq \mathbb{E}_{\mathbf{x}\sim\hat{\mathbb{P}}_s} [f_w(\mathbf{x})] - \frac{1}{N}\sum_i \left[ \mathbb{E}_{\mathbf{x}\sim\hat{\mathbb{P}}_{\theta_i}} [f_w(\mathbf{x})] \right]$$

$$= \mathbb{E}_{\mathbf{x}\sim\mathbb{P}_s} [f_w(\mathbf{x})] - \frac{1}{N}\sum_i \left[ \mathbb{E}_{\mathbf{x}\sim\mathbb{P}_{\theta_i}} [f_w(\mathbf{x})] \right] - \left( \mathbb{E}_{\mathbf{x}\sim\mathbb{P}_s} [f_w(\mathbf{x})] - \mathbb{E}_{\mathbf{x}\sim\hat{\mathbb{P}}_s} [f_w(\mathbf{x})] \right)$$

$$- \frac{1}{N}\sum_i \left[ \mathbb{E}_{\mathbf{x}\sim\hat{\mathbb{P}}_{\theta_i}} [f_w(\mathbf{x})] - \mathbb{E}_{\mathbf{x}\sim\mathbb{P}_{\theta_i}} [f_w(\mathbf{x})] \right]$$

$$\geq W(\mathbb{P}_s, \mathbb{P}_{\theta_1}, \ldots, \mathbb{P}_{\theta_N}) - \epsilon.$$

Similarly, $W(\mathbb{P}_s, \mathbb{P}_{\theta_1}, \ldots, \mathbb{P}_{\theta_N}) \geq W\left( \hat{\mathbb{P}}_s, \hat{\mathbb{P}}_{\theta_1}, \ldots, \hat{\mathbb{P}}_{\theta_N} \right) - \epsilon$. Therefore, when the number of sample in each domain satisfying $n \geq \frac{C\kappa\Delta^2 \log(L\kappa/\epsilon)}{\epsilon^2}$, then the following satisfies with the probability at least $1 - \exp(-\kappa)$,

$$\left| W\left( \hat{\mathbb{P}}_s, \hat{\mathbb{P}}_{\theta_1}, \ldots, \hat{\mathbb{P}}_{\theta_N} \right) - W(\mathbb{P}_s, \mathbb{P}_{\theta_1}, \ldots, \mathbb{P}_{\theta_N}) \right| \leq \epsilon.$$

We conclude the proof.   $\square$

## G   Proof of Lemma 1

**Lemma 1** (**Constraints relaxation**) *If the cost function $c(\cdot)$ is measured by $\ell_2$ norm, then there exists a constant $L_f \geq 1$ such that discriminator $f$ satisfies the following constraint:*

$$\sum_i \frac{\left| f(\mathbf{x}) - f\left(\hat{\mathbf{x}}^{(i)}\right) \right|}{\left\| \mathbf{x} - \hat{\mathbf{x}}^{(i)} \right\|} \leq L_f. \tag{49}$$

**Proof**   When the inequality constraints in Problem IV is satisfied, and without loss of generality, we assume that $\frac{1}{N} \sum \left| f(\mathbf{x}) - f\left(\mathbf{x}^{(i)}\right) \right| \leq c(\mathbf{x}^{(0)}, \ldots, \mathbf{x}^{(N)})$. Let $c := c(\mathbf{x}^{(0)}, \ldots, \mathbf{x}^{(N)})$, we have

$$\frac{1}{Nc} \min_i \left\| \mathbf{x} - \hat{\mathbf{x}}^{(i)} \right\| \sum_i \frac{\left| f(\mathbf{x}) - f\left(\hat{\mathbf{x}}^{(i)}\right) \right|}{\left\| \mathbf{x} - \hat{\mathbf{x}}^{(i)} \right\|} \leq \frac{1}{Nc} \sum_i \frac{\left\| \mathbf{x} - \hat{\mathbf{x}}^{(i)} \right\| \left| f(\mathbf{x}) - f\left(\hat{\mathbf{x}}^{(i)}\right) \right|}{\left\| \mathbf{x} - \hat{\mathbf{x}}^{(i)} \right\|} \leq 1.$$

Let $L_f = Nc / \min_i \left\| \mathbf{x} - \hat{\mathbf{x}}^{(i)} \right\| \geq 1$, we conclude the proof.   $\square$

In Lemma 1, the constant $L_f$ is related to the cost function $c$. In this sense, it captures the dependency among domains.

## H   Discussions on Lipschitz Condition

From the following proposition, the assumption that the potential function is Lipschitz continuous is strong to enforce the inequality constraints. It would cause misleading results for our problem setting.

**Proposition 1** *If the potential function is Lipschitz continuous, and the cost function is defined as*

$$c\left(\mathbf{x}, \mathbf{x}^{(1)} \ldots, \mathbf{x}^{(N)}\right) = \sum_{i \in [N]} \left\| \mathbf{x} - \mathbf{x}^i \right\|, \tag{50}$$

*then the potential function must satisfy the inequality constraints, i.e.,*

$$\frac{1}{N} \sum_{i \in [N]} \left| f(\mathbf{x}) - f\left(\mathbf{x}^{(i)}\right) \right| \leq c\left(\mathbf{x}, \mathbf{x}^{(1)} \ldots, \mathbf{x}^{(N)}\right). \tag{51}$$

**Proof**   If the potential function is 1-Lipschitz continuous, *i.e.*,

$$\left| f(\mathbf{x}) - f\left(\mathbf{x}^i\right) \right| \leq \left\| \mathbf{x} - \mathbf{x}^i \right\|, \quad i \in [N]. \tag{52}$$

Then, based on the definition of the potential and for all variables, we have

$$\frac{1}{N} \sum_{i \in [N]} \left| f(\mathbf{x}) - f\left(\mathbf{x}^{(i)}\right) \right| \leq \sum_{i \in [N]} \left\| \mathbf{x} - \mathbf{x}^i \right\|. \tag{53}$$

$\square$

# I Comparisons with GAN Methods

## I.1 Differences between MWGAN and WGAN

In this paper, the proposed MWGAN essentially differs from WGAN even when $\lambda_i^+ = 1/N$: **1)** MWGAN considers and incorporates multi-domain correlations into the inequality constraints to improve the **image translation** performance. WGAN focuses on **image generation tasks** and cannot directly deal with multi-domain correlations. **2)** The objectives of two methods are different in the formulation. **3)** In the algorithm, MWGAN uses gradient penalty to deal with inequality constraints; while WGAN relies on the weight clipping.

## I.2 Comparisons with Image-to-image Translation Methods

CycleGAN [20] is a two-domain translation method, but it can be used in the multi-domain image translation task. It means that CycleGAN needs to learn multiple two-domain translation tasks. Moreover, CycleGAN performs well on the unbalanced translation task, because it independently optimizes multiple individual networks for the multi-domain image translation task. StarGAN [6] and UFDN [13] are multi-domain image translation methods, however, they may not exploit multi-domain correlations to achieve good performance on the unbalanced translation task.

Table 4: Comparisons with image-to-image translation methods.

| Method | Unpaired data | Multiple domains | Multi-domain correlations | Unbalanced translation task |
|--------|:---:|:---:|:---:|:---:|
| CycleGAN | ✓ | ✗ | ✗ | ✓ |
| StarGAN | ✓ | ✓ | ✗ | ✗ |
| UFDN | ✓ | ✓ | ✗ | ✗ |
| MWGAN | ✓ | ✓ | ✓ | ✓ |

**Difference between MWGAN and StarGAN.** The adversarial learning of MWGAN is different from StarGAN. Specifically, MWGAN cannot be interpreted as distribution matching between source and a mixture of target distributions. Let $\bar{\mathbb{P}}_\theta$ be a mixture distribution over $(\mathbb{P}_{\theta_1}, \ldots, \mathbb{P}_{\theta_N})$. Note that $\bar{\mathbb{P}}_\theta$ is related to the batch size. When the batch size is too small, then $\bar{\mathbb{P}}_\theta$ cannot guarantee to contain all domains. StarGAN minimizes the following optimization problem:

$$\max_f \mathbb{E}_{\mathbf{x} \sim \hat{\mathbb{P}}_s}[f(\mathbf{x})] - \mathbb{E}_{\hat{\mathbf{x}} \sim \bar{\mathbb{P}}_\theta}[f(\hat{\mathbf{x}})]. \tag{54}$$

In contrast, MWGAN minimizes the following optimization problem,

$$\max_f \mathbb{E}_{\mathbf{x} \sim \hat{\mathbb{P}}_s}[f(\mathbf{x})] - \sum_i \lambda_i^+ \mathbb{E}_{\hat{\mathbf{x}} \sim \hat{\mathbb{P}}_{\theta_i}}[f(\hat{\mathbf{x}})]. \tag{55}$$

When $\lambda_i^+ = 1/N$ and $\bar{\mathbb{P}}_\theta$ is uniformly drawn from every target generated distribution, the objective (54) is equivalent to the objective (55). However, when $\lambda_i^+ \neq 1/N$ and the batch size is small, the objective (54) is not equivalent to the objective (55). Besides, the inequality constraints in MWGAN are related to the correlation among all domains, while StarGAN only considers the source domain and certain target domain. Therefore, the adversarial learning of MWGAN is different from StarGAN.

## J Effectiveness of One Potential Function

### J.1 Comparisons between One and Multiple Potential Functions

We focus on multiple marginal matching, where multiple target domains often contain cross-domain correlations. Thus, a shared potential function helps to exploit cross-domain correlations to improve performance (see results in Table 5 on the Edge→CelebA task). Second, training a shared function using entire data of all domains is much easier than training $N+1$ potentials (one per domain).

Table 5: Comparisons of shared and $N+1$ potentials in terms of FID.

| Method | Black hair | Blond hair | Brown hair |
|---|---|---|---|
| $N+1$ potentials | 245.25 | 289.56 | 303.04 |
| One potential | 33.81 | 51.87 | 35.24 |

### J.2 Weight Setting and Performance vs #domains ($N$)

With $\lambda_i^+ = 1/N$, each generator provides equal gradient feedbacks in each target domain and helps to exploit cross-domain correlations in adversarial learning. We apply MWGAN on the Edge→CelebA translation task with different $N$. From Table 6, more domains help to improve the performance in terms of FID by exploiting cross-domain correlations.

Table 6: Performance vs #domains in FID.

| $N$ | 2 | 3 | 4 | 5 |
|---|---|---|---|---|
| FID | 58.61 | 38.31 | 33.81 | 32.43 |

# K  Toy Dataset

## K.1  7 Gaussian Distributions

In the first row of Figure 2, we generate 7 Gaussian distributions as the real data distribution, where the center of initial distribution (green) is $(0,0)$, and the centers of 6 target distributions (red) are $(3/2,0)$, $(-3/2,0)$, $(3/4, 3\sqrt{3}/4)$, $(-3/4, 3\sqrt{3}/4)$, $(3/4, -3\sqrt{3}/4)$ and $(-3/4, -3\sqrt{3}/4)$. For each Gaussian distribution, the variance is $0.04$, and we generate 256 samples. The synthetic data distribution (orange) is generated from the Gaussian centered at $(0,0)$.

## K.2  1 Gaussian and 6 Uniform Distributions

In the second row of Figure 2, we generate 1 Gaussian distribution and 6 uniform distributions as the real data distributions, where the center of initial distribution (green) is also $(0,0)$, and the centers of 6 uniform distributions (red) are $(3/2,0)$, $(-3/2,0)$, $(3/4, 3\sqrt{3}/4)$, $(-3/4, 3\sqrt{3}/4)$, $(3/4, -3\sqrt{3}/4)$ and $(-3/4, -3\sqrt{3}/4)$. For each uniform distribution, we generate 256 samples in a square around the center (length is 0.4). The synthetic data distribution (orange) is generated from the Gaussian centered at $(0,0)$.

## K.3  Toy Experiment Settings

We use fully connected neural network architecture for all methods. The generator contains 3 hidden layers with 512 units followed by ReLU. The discriminator contains 2 hidden layers with 512 units followed by ReLU. We use Adam as the optimizer with $\beta_1 = 0.5$ and $\beta_2 = 0.999$ and the learning rate of all methods is set to 0.0001. The hyper-parameters follow the default setting of these methods.

# L  Details of Classification on CelebA

In the facial attribute translation experiment (In Section 6.5 of the main submission), we train a classifier on CelebA to obtain a near-perfect accuracy, and test on blond hair, eyeglasses, mustache and pale skin to obtain classifier accuracy of 99.62%, 99.94%, 99.76% and 97.96%, respectively. In the same way, we use this classifier to test on synthesized single and multiple attributes for the considered methods.

## M    More Evaluations with Amazon Mechanical Turk (AMT)

For more quantitative evaluations, we conduct a perceptual evaluation using AMT to assess the performance on the Edge→CelebA translation task, following the settings of StarGAN [6]. From Table 7, MWGAN wins significant majority votes for the best perceptual realism, quality and transferred attributes for all facial attributes.

Table 7: AMT perceptual evaluation for each attribute.

| Method | Black hair | Blond hair | Brown hair |
|---|---|---|---|
| CycleGAN | 9.7% | 5.7% | 9.0% |
| UFDN | 13.2% | 15.8% | 12.9% |
| StarGAN | 16.0% | 21.9% | 19.4% |
| MWGAN | **61.1%** | **56.6%** | **58.7%** |

## N    Influences of Inner-domain and Inter-domain Constraints

In this section, we evaluate the influences of inner-domain constraints and inter-domain constraints on the edge→celebA task, respectively. Specifically, we compare the FID values with different $\alpha$ (inner-domain constraint weights) and different $\tau$ (inter-domain constraint weights). The value of $\alpha$ and $\tau$ is selected among [0, 0.1, 1, 10, 100]. Each experiment only evaluates one constraint and fix other parameters. To evaluate the influence of $\alpha$, we empirically set $\tau = 10$. Otherwise, we set $\alpha = 10$ to evaluate the influence of $\tau$. The results are shown in Tables 8 and 9.

Specifically, when $\alpha$=0 or $\tau$=0, MWGAN obtains the worst performance. In other words, when we abandon any one of the inner or inter-domain constraints, we cannot achieve a satisfactory result. This demonstrates the effectiveness of both constraints. Besides, MWGAN achieves the best performance when setting both weights to 10. This means that when setting some reasonable constraint weights, we can achieve a better trade-off between the optimization objective and constraints, and thus obtain better performance.

Table 8: Influence of $\alpha$ for the inner-domain constraint in terms of FID.

| $\alpha$ | Black hair | Blond hair | Brown hair |
|---|---|---|---|
| 0 | 316.41 | 334.08 | 325.31 |
| 0.1 | 263.85 | 317.89 | 300.79 |
| 1 | 109.65 | 109.48 | 136.97 |
| 10 | **33.81** | **51.87** | **35.24** |
| 100 | 55.96 | 71.60 | 66.17 |

Table 9: Influence of $\tau$ for the inter-domain constraint in terms of FID.

| $\tau$ | Black hair | Blond hair | Brown hair |
|---|---|---|---|
| 0 | 392.87 | 360.17 | 346.16 |
| 0.1 | 276.07 | 328.16 | 337.11 |
| 1 | 90.75 | 87.99 | 93.18 |
| 10 | **33.81** | **51.87** | **35.24** |
| 100 | 54.30 | 56.44 | 48.03 |

## O    Influences of the Parameter $L_f$

In this section, we evaluate the influences of the parameter $L_f$ on the edge→celebA task. Specifically, we compare the FID values with different $L_f$ in the inter-domain constraints. The value of $L_f$ is selected among [1, 3, 10, 50], where 3 is the number of domains. Each experiment only evaluates one constraint and fix other parameters. The results are shown in Table 10.

In Tables 10, MWGAN achieves the best performance when setting both weights to 3. This means that when setting some reasonable constants, we can achieve a better gradient penalty between the source and each target domain, and thus obtain better performance by exploiting the cross-domain correlations.

Table 10: Influence of the parameter $L_f$ for the domain constraint in terms of FID.

| $L_f$ | Black hair | Blond hair | Brown hair |
|---|---|---|---|
| 1 | 64.17 | 55.98 | 49.19 |
| 3 | **33.81** | **51.87** | **35.24** |
| 10 | 44.12 | 52.46 | 43.64 |
| 50 | 79.89 | 91.07 | 79.50 |

## P  Network Architecture and More Implementation Details

**Network architecture.**    The classifier $\phi$ shares the same structure except for the output layer with $f$. The network architectures of the discriminator and generators of MWGAN are shown in Tables 11 and 12. We split each generator to an encoder and a decoder, where all generators share the same encoder but with different decoders. For the encoder and decoder network, instead of using batch normalization [10, 11], we use instance normalization in all layers except the last output layer of the decoder. For the discriminator network, we use PatchGAN network which is made up of fully convolutional networks, and we use Leaky ReLU with a negative slope of 0.01. We use the following abbreviations: $h$: the width size of input image, $w$: the height size of input image, $n_d$: the number of transferred domains(exclude source domain), N: the number of output channels, K: kernel size, S: stride size, P: padding size, IN: instance normalization.

**More implementation details.**    For Loss (5), we use mean square loss and cross-entropy loss for the balanced and imbalanced translation task, respectively. In the experiments, we find that introducing an identity mapping loss [20] helps improve the quality of generated images on the facial attribute translation task. Specifically, the identity mapping loss is defined as: $\mathcal{L}_{\text{idt}}(g_i) = \mathbb{E}_{\mathbf{x}\sim\hat{\mathbb{P}}_{t_i}}\left[\|g_i(\mathbf{x}) - \mathbf{x}\|_1\right]$, where $\hat{\mathbb{P}}_{t_i}$ is an empirical distribution in the $i$-th target domain. We use the identity mapping loss for all target domains.

Table 11: Generator network architecture.

| Encoder | | |
|---|---|---|
| Part | Input $\rightarrow$ Output shape | Layer information |
| Down-sampling | $(h, w, 3)\rightarrow(h, w, 64)$ | CONV-(N64, K7x7, S1, P3), IN, ReLU |
| | $(h, w, 64)\rightarrow(\frac{h}{2}, \frac{w}{2}, 128)$ | CONV-(N128, K4x4, S2, P1), IN, ReLU |
| | $(\frac{h}{2}, \frac{w}{2}, 128)\rightarrow(\frac{h}{4}, \frac{w}{4}, 256)$ | CONV-(N256, K4x4, S2, P1), IN, ReLU |
| Bottleneck | $(\frac{h}{4}, \frac{w}{4}, 256)\rightarrow(\frac{h}{4}, \frac{w}{4}, 256)$ | Residual Block: CONV-(N256, K3x3, S1, P1), IN, ReLU |
| | $(\frac{h}{4}, \frac{w}{4}, 256)\rightarrow(\frac{h}{4}, \frac{w}{4}, 256)$ | Residual Block: CONV-(N256, K3x3, S1, P1), IN, ReLU |
| | $(\frac{h}{4}, \frac{w}{4}, 256)\rightarrow(\frac{h}{4}, \frac{w}{4}, 256)$ | Residual Block: CONV-(N256, K3x3, S1, P1), IN, ReLU |
| Decoder | | |
| Bottleneck | $(\frac{h}{4}, \frac{w}{4}, 256)\rightarrow(\frac{h}{4}, \frac{w}{4}, 256)$ | Residual Block: CONV-(N256, K3x3, S1, P1), IN, ReLU |
| | $(\frac{h}{4}, \frac{w}{4}, 256)\rightarrow(\frac{h}{4}, \frac{w}{4}, 256)$ | Residual Block: CONV-(N256, K3x3, S1, P1), IN, ReLU |
| | $(\frac{h}{4}, \frac{w}{4}, 256)\rightarrow(\frac{h}{4}, \frac{w}{4}, 256)$ | Residual Block: CONV-(N256, K3x3, S1, P1), IN, ReLU |
| Up-sampling | $(\frac{h}{4}, \frac{w}{4}, 256)\rightarrow(\frac{h}{2}, \frac{w}{2}, 128)$ | DECONV-(N128, K4x4, S2, P1), IN, ReLU |
| | $(\frac{h}{2}, \frac{w}{2}, 128)\rightarrow(h, w, 64)$ | DECONV-(N64, K4x4, S2, P1), IN, ReLU |
| | $(h, w, 64)\rightarrow(h, w, 3)$ | CONV-(N3, K7x7, S1, P3), Tanh |

Table 12: Discriminator network architecture.

| Layer | Input $\rightarrow$ Output shape | Layer information |
|---|---|---|
| Input Layer | $(h, w, 3)\rightarrow(\frac{h}{2}, \frac{w}{2}, 64)$ | CONV-(N64, K4x4, S2, P1), Leaky ReLU |
| Hidden Layer | $(\frac{h}{2}, \frac{w}{2}, 64)\rightarrow(\frac{h}{4}, \frac{w}{4}, 128)$ | CONV-(N128, K4x4, S2, P1), Leaky ReLU |
| Hidden Layer | $(\frac{h}{4}, \frac{w}{4}, 128)\rightarrow(\frac{h}{8}, \frac{w}{8}, 256)$ | CONV-(N256, K4x4, S2, P1), Leaky ReLU |
| Hidden Layer | $(\frac{h}{8}, \frac{w}{8}, 256)\rightarrow(\frac{h}{16}, \frac{w}{16}, 512)$ | CONV-(N512, K4x4, S2, P1), Leaky ReLU |
| Hidden Layer | $(\frac{h}{16}, \frac{w}{16}, 512)\rightarrow(\frac{h}{32}, \frac{w}{32}, 1024)$ | CONV-(N1024, K4x4, S2, P1), Leaky ReLU |
| Hidden Layer | $(\frac{h}{32}, \frac{w}{32}, 1024)\rightarrow(\frac{h}{64}, \frac{w}{64}, 2048)$ | CONV-(N2048, K4x4, S2, P1), Leaky ReLU |
| Output layer ($f$) | $(\frac{h}{64}, \frac{w}{64}, 2048)\rightarrow(\frac{h}{64}, \frac{w}{64}, 1)$ | CONV-(N1, K3x3, S1, P1) |
| Output layer ($\phi$) | $(\frac{h}{64}, \frac{w}{64}, 2048)\rightarrow(1, 1, n_d)$ | CONV-(N($n_d$), K$\frac{h}{64}$x$\frac{w}{64}$, S1, P0) |

# Q    Additional Qualitative Results

## Q.1    Results on CelebA

| Input | Blond hair | Eyeglasses | Mustache | Pale skin | B+E | B+M | B+E+M |

Figure 6: Single and multiple attribute translation results on CelebA.

## Q.2 Results on Edge→CelebA

Figure 7: Translation results from edge images to CelebA.

## Q.3 Results on Painting Translation

Figure 8: Translation results from real-world images to painting images.