[Reviews · NeurIPS 2019]

Reviewer 1



Bases on the optimal transportation theory, the authors have developed a new approach to tackle the multiple marginal matching problem. The authors have provided details to derive tractable objective function, which can be formulated as a GAN problem eventually. Theoretical analysis on the generalization of the method has been conducted. Experiments on both toy and real-world data are helpful to justify the effectiveness of the method. In Section 4.1, the authors linked potential functions in multiple domains using a uniform potential function with different weights. This assumption seems to be a bit strong. Some more explanations are necessary to justify the reasonability of this formulation. In Problem IV, the authors state that 1/N can be taken as a default value for lambda. If so, the problem will be nearly the same with the objective function of classical WGAN. Further considering the generators for different domain, the whole process of the algorithm may not show significant difference from WGAN, except that more than one generator has been used. In addition, the correlation between multiple domains seems to be investigated in a very straightforward way by assuming that a shared discriminator (in WGAN) for multiple domains. It is therefore unclear whether this simply approach indeed is helpful for exploring the correlation. It is very interesting to discuss the generalization in the paper. According to definition 1, the generalization is defined over training sample. How about the generalization over unseen test examples? ------------------------------------------------------------------------------ The authors addressed my concerns on the technical details in the rebuttal. The proposed algorithm is theoretically motivated and has shown performance advantages in experiments. This paper is interesting to me, and I would like to vote for an acceptance.

Reviewer 2



Multi Marginal Wasserstein GAN's goal is to match a source domain distribution to multiple target domain distributions. While dedicated GAN frameworks exist, as noted by the authors (CycleGAN, StarGAN, …), their generated samples suffer from blurriness, especially when resulting from multiple targets. Moreover, the main statement of this work is that MWGAN is theoretically motivated, unlike previous works. Computing the multi marginal Wasserstein distance between several domains is intractable in its primal form. Thus, as proposed in WGAN, the authors express their problem in the dual form, resulting in equation 1. Since, the dual formulation is a maximization problem under infinite constraint, which remains intractable, the authors simplify the problem by only considering it on its empirical version (equation 2). Eventually, the authors argue that if the potential functions can be expressed by a unique function up to a multiplicative constant, then they can simplify Problem III, that they finally enounce in Problem IV as the Multi Marginal Wasserstein GAN (MWGAN). When it comes to the training of MWGAN, the framework requires two additional terms as presented in Algorithm 1: -inner domain constraints: a classifier to constrain generator number i to sample from the i-th domain -inter-domain constraints: when the generators have not converged yet, it may jeopardize the training to constraint the generated samples to follow the inequality constraints or Problem III. Then the authors propose a softer version that balances the loss function. Something that is not clear is that it seems that the authors claim to solve the multi marginal Wasserstein distance, which is theoretically wrong as they make hard approximations on the family of potential functions, which may mislead the reader. Moreover, there is no discussion on cases where this approximation may be true or any discussions on the tightness of this approximation. Nevertheless, the authors rely on the section Theoretical Discussion, to promote the generalization ability of their method with enough training data. Again, I am not sure that those bounds are true for their approximation and I would appreciate some clarifications. When it comes to the experimental sections, results have been conducted thoroughly on several datasets used two criterions FID (which is not useful in the multi-target transfer) and a classifier trained on the ground truth data to recognize the domains. An AMT perceptual evaluation as conducted by StarGAN would have been interested also. Eventually, the results and their illustrations are promising. I would be curious about the composite generator: when applying multiple attributes, how do you pick the order of compositions (I expect it not to commute). In a Nutshell here is the list of the main pros and cons of this work: - pros: The experiments are state of the art with high quality generated samples for multiple attributes - cons: the authors overly claim their theoretical guarantees thanks to the multi marginal Wasserstein distance, without any analyses of the tightness of their approximations on the potential functions.

Reviewer 3



Originality:This paper solves the multiple marginal mapping problem by defining a multi-marginal Wasserstein algorithm firstly. Quality: The whole structure of this work is consistent in general. Under a specific condition, the paper gives the sound technically analysis, contains the equivalence of solutions, and the generalization analysis. And the theoretical analysis and the empirical experiment results to support for the proposed algorithms. Clarity:The paper is written clearly and easy to follow. Significance: This work makes a moderate advance for M3 problem. Under a specific and rigorous condition, the authors done an adequate work theoretically and experimentally. The only problem is that how can real problems satisfy the condition. Some concerns: 1)The whole work stand on a condition that a shared potential function is sufficient for problem 1 in paper. But authors just use the experimental result in appendix I to show the practicability in some real-world tasks. It seems weak. 2)In theorem 1, there is a key assumption which states “if (f_0, \cdots, f_N) and (\lambda_0f , \cdots, \labmbda_N f ) are solutions to problem 1”. How can you verify this assumption? The key question I want to known is, why you can replace N different functions with a unified function f and N constant factor? Or within what distance among the multiple domain, you can do such replace. 3)Toy data experiment. Can you explain more detailly about how to understand the Figure2 especially the value surface of the discriminator.

[Author Response · NeurIPS 2019]

We thank all reviewers for the recognition on the **novelty** and **quality** of our paper: "this work is theoretically motivated,
unlike previous works", "state-of-the-art results with high quality ..."(**R2**), "very interesting to discuss the generalization"
(**R1**), and "makes a moderate advance for $M^3$ problem"(**R3**). We first answer a general concern from reviewers.

**General Response:**

**G1. Why use a shared potential function?** We address the concern with both **Empirical** and **Theoretical** evidences.

**(1) Motivation and empirical justification.** We use the shared potential function to exploit the cross-domain correla-
tions for $M^3$ problem. From Table 6 in Appendix J, more domains indeed help to improve the performance.

**(2) Theoretical justification.** It is valid to use a shared potential function to replace $N$ ones. In fact, we can prove
that the optimal objective of Problem II (with $N$ potential functions) is close to Problem III (with a shared potential
function) under mild conditions over $\{\lambda_i\}$ and the cost function. In an extreme case, if $N+1$ domains have overlapped
samples, the optimal objectives of Problems II and III are equal. These verify the **assumption** in Theorem 1.

**Proof sketch:** We define the cost function as $c(\mathbf{x}^{(0)}, \dots, \mathbf{x}^{(N)}) = \sum_{i \neq j} d(\mathbf{x}^{(i)}, \mathbf{x}^{(j)})$, where $d(\cdot, \cdot)$ is a distance function
of two samples and $\mathbf{x}^{(i)}$ is a sample in the $i$-th domain. The proof can be adapted from the proof of Theorem 3.3 in [24].
For $N{=}1$ (*i.e.*, two domains), if $d(\cdot, \cdot)$ satisfies the triangle inequality, the optimal objective of Problems II and III are
equal [24]. Similarly, for $N{\geq}2$, the equivalence holds when $d(\cdot, \cdot)$ satisfies the triangle inequality and $\sum_i \lambda_i{=}0$. Let
$f^*$ be an optimizer of Problem III. We first prove $\lambda_0 f^*(\mathbf{x}^{(0)}){=}\inf\{c(\mathbf{x}^{(0)}, \mathbf{x}^{(1)}, \dots, \mathbf{x}^{(N)}) - \sum_{j \in [N]} \lambda_j f^*(\mathbf{x}^{(j)})\}$. Let $f_i^*$
be optimal solutions to Problem II and $(f^c)^*$ be the $c$-conjugate function. If $N+1$ domains have overlapped samples
$\mathbf{x}_{k_i}^{(i)} \in \mathcal{X}^{(i)}$ (see [24]), we can prove $(f^c)^*(\mathbf{x}_{k_0}^{(0)}){=}f_i^*(\mathbf{x}_{k_i}^{(i)}), i{\in}[N]$. Last, we prove any optimal solution to Problem II (resp.
III) is a feasible solution to Problem III (resp. II). Then, we conclude the optimal objectives of Problems II and III are
equal. For more general cases, we instead prove that the optimal objectives of Problems II and III can be arbitrarily
close when multiple domains are very close to each other. We leave the complete proofs in the revised paper. □

**To Reviewer #1 (R1):**

**Q1. More explanations of a shared potential function & Can it exploit correlations?** See **General Response G1**.

**Q2. Differences of MWGAN from WGAN [3].** MWGAN essentially differs from WGAN even when $\lambda_i^+ = 1/N$:
**1)** MWGAN considers and incorporates multi-domain correlations into the inequality constraints to improve the **image
translation** performance. WGAN focuses on **image generation tasks** and cannot directly deal with multi-domain
correlations. **2)** The objectives of two methods are different in the formulation. **3)** In the algorithm, MWGAN uses
gradient penalty to deal with inequality constraints; while WGAN relies on weight clipping.

**Q3. Generalization on unseen test samples.** Our definition on generalization has considered testing samples (which
is similar to [30]). Specifically, in Definition 1, $\mathbb{P}_s$ denotes the probability distribution of unseen source samples.

**To Reviewer #2 (R2):**

**Q1. Theoretical justification of approximation on the potential function.** Please refer to **General Response G1**.

**Q2. More evaluations with Amazon Mechanical Turk (AMT).**
We conduct a perceptual evaluation using AMT to assess the
performance on the Edge→CelebA translation task, following
the settings of StarGAN [6]. From Table A, MWGAN wins
significant majority votes for the best perceptual realism, quality
and transferred attributes for all facial attributes.

Table A: AMT perceptual evaluation for each attribute.

| Method | Black hair | Blond hair | Brown hair |
|---|---|---|---|
| CycleGAN | 9.7% | 5.7% | 9.0% |
| UFDN | 13.2% | 15.8% | 12.9% |
| StarGAN | 16.0% | 21.9% | 19.4% |
| MWGAN | **61.1%** | **56.6%** | **58.7%** |

**Q3. Order of compositions.** We generate attributes with order {Blond hair, Eyeglasses, Mustache and Pale skin},
which works well. The order has a slight impact on the performance. We will include relevant results and discussions.

**To Reviewer #3 (R3):**

**Q1. Empirical and Theoretical sufficiency of a shared potential function.** Please refer to **General Response G1**.

**Q2. Metrics of domain similarity and its relation to conditions of the shared potential function.** The domain
similarity/correlation indeed is very critical for our method and theoretical analysis. We start to measure the distance
among multiple domains with multi-marginal Wasserstein distance, which however is hard to compute. We thus propose
a new feasible dual formulation. From **General Response G1**, if domains are close enough upon sample distances
$d(\cdot, \cdot)$, we can use a shared potential function. Nevertheless, in many real problems (*e.g.*, the image translation task),
different domains indeed have high correlations, where our method achieved promising performance (See Table A).

**Q3. How to understand Fig. 2?** Fig. 2 is to show the distribution matching abilities of various methods. The value
surface, which depicts the output of the discriminator, is widely used in [14, 24]. More discussions will be included.

[Meta-Review · NeurIPS 2019]

The paper introduces a new, theoretically motivated approach to multi-margin matching. The dual formulation in terms of a shared potential is quite elegant, and experiments are promising. Please take reviewer comments into account when preparing the final version of the paper.